# Computing the Sound–Sense Harmony: A Case Study of William Shakespeare's Sonnets and Francis Webb's Most Popular Poems

**Rodolfo Delmonte** 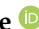

**Abstract:** Poetic devices implicitly work towards inducing the reader to associate intended and expressed meaning to the sounds of the poem. In turn, sounds may be organized a priori into categories and assigned presumed meaning as suggested by traditional literary studies. To compute the degree of harmony and disharmony, I have automatically extracted the sound grids of all the sonnets by William Shakespeare and have combined them with the themes expressed by their contents. In a first experiment, sounds have been associated with lexically and semantically based sentiment analysis, obtaining an 80% of agreement. In a second experiment, sentiment analysis has been substituted by Appraisal Theory, thus obtaining a more fine-grained interpretation that combines dis-harmony with irony. The computation for Francis Webb is based on his most popular 100 poems and combines automatic semantically and lexically based sentiment analysis with sound grids. The results produce visual maps that clearly separate poems into three clusters: negative harmony, positive harmony and disharmony, where the latter instantiates the need by the poet to encompass the opposites in a desperate attempt to reconcile them. Shakespeare and Webb have been chosen to prove the applicability of the method proposed in general contexts of poetry, exhibiting the widest possible gap at all linguistic and poetic levels.

**Keywords:** specialized NLP system for poetry; automatic poetic analysis; visualization of linguistic and poetic content; Sound–Sense matching algorithm; phonetic and phonological analysis; automatic lexical and semantic sentiment analysis; computing irony; appraisal theory framework



## 1. Introduction

In this article, I will propose a totally new technique to assess and appreciate poetry, the *Algorithm for Sound and Sense Harmony (henceforth ASSH)*. The main tenet of this paper is the existence of a hidden and systematic plan by important poets like Shakespeare and Webb to organize rhyming structures in accordance with a principle of overall ASSH. What is meant here by "Sound Harmony" is the presence of rhymes whose sound—the stressed vowel that is dominant—belongs to the four sound classes that may comprise all vowel sounds, phonologically speaking, i.e., low, mid, high-front, high-back, or part of them. In addition, the "Sound Harmony" is composed with Sense to make up the ASSH, where the choice of sounds reflects the contents of the poem, as it may be represented by main topics, intended meaning and overall sentiment. The same argument is presented for the presence of the three main classes of consonants, i.e., continuants, sonorants, obstruents and their partition into voiced vs. unvoiced. The choice to favor the presence of one class vs. another is to be interpreted as a way to highlight sense-related choices of words that will either accompany or contrast with Sounds. In particular, we associate different mood—following traditional judgements—to vowels and consonants according to their class, as follows:

1. Low and mid vowels evoke a sense of brightness, peace and serenity;

2. High, front and back vowels evoke a sense of surprise, seriousness, rigor and gravity;
3. Obstruent and unvoiced consonants evoke a sense of harshness and severity;
4. Sonorant and continuant consonants evoke a sense of pleasure, softness and lightness.

Classes 1 and 4 will be regarded in the same area of positive thinking, while classes 2 and 3 will more naturally be accompanied by negative sentiment. Of course, it may be the case that crossed matches with classes belonging to opposite types will take place more or less frequently, indicating the need to reconcile opposite feelings in the same poem. This is what happens in both Shakespeare's and Webb's poems, as will be shown in the sections below.

It is important to highlight the role of sounds in poetry, which is paramount for the creation of poetic and rhetoric devices. Rhyme, alliterations, assonances and consonances may contribute secondary and, in some cases, primary additional meaning by allowing words which are not otherwise syntactically or semantically related to share part if not all of their meaning by means of metaphors and other similar devices. Thus, most of the difficult work of every poet is devoted to the choice of the appropriate word to use for rhyming purposes, mainly, but also for the other important devices mentioned above.

In the case of Shakespeare, for the majority of the sonnets, he took care of choosing words for the rhymes contributing sounds to the four varieties, thus producing a highly varied sound harmony. We will discuss this in the sections below, paying attention to associate choice of one class vs. another, with choice of specific themes or words. This important feature of the sonnets has never been noticed by literary critics in the past. Reasons for this apparent lack of attention may be imputed to the existence of two seemingly hindering factors: a former factor is the use of words which had a double pronunciation at the time, as for instance LOVE which could be pronounced as MOVE in addition to its current pronunciation. The latter factor regards the existence of a high—in comparison with other poets of the same Elizabethan period—percentage of a variable we call Rhyme Repetition Rate (TripleR), which indicates the use of the same "head" word—i.e., the rhyming word that precedes the alternate rhyme scheme—or sometimes the same couple of words.

The use of mood and related colours associated with sound in poetry has a long tradition. Rimbaud composed a poem devoted to "Vowels", where colours were associated with each of the main five vowels. Roman Jakobson [1,2] and Mazzeo [3] wrote extensively about the connection between sound and colour in a number of papers. Fónagy [4] wrote an article in which he explicitly connected the use of certain types of consonant sounds associated with certain moods: unvoiced and obstruent consonants are associated with aggressive mood; sonorants with tender moods. Macdermott [5] identified a specific quality associated with "dark" vowels, i.e., back vowels, that of being linked with dark colours, mystic obscurity, hatred and struggle. As a result, we are using darker colours to highlight back and front vowels as opposed to low and middle vowels, the latter with light colours. The same applies to representing unvoiced and obstruent consonants as opposed to voiced and sonorants. But as Tsur (see [6], p. 15) notes, this sound–colour association with mood or attitude has no real significance without a link to semantics.

In one of the visual outputs produced by our system, SPARSAR—presented in a section below, the Semantic Relational View, we are using dark colours for *concrete* referents vs. *abstract* ones [7] with lighter colours; and dark colours also for *negatively* marked words as opposed to *positively* marked ones with lighter colours. The same strategy applies to other poetic maps: this technique has certainly the good quality of highlighting opposing differences at some level of abstraction. Our approach is not comparable to work by Saif Mohammad [8], where colours are associated with words on the basis of what their mental image may suggest to the mind of annotators hired via Mechanical Turk (Amazon Mechanical Turk (MTurk) is a crowdsourcing marketplace that makes it easier for individuals and businesses to outsource their processes and jobs). The resource only contains word–colour association for some 12,000 entries over the 27 K items listed. It is, however, comparable to a long list of other attempts at depicting phonetic differences in

poems as will be discussed further on. With this experiment, I intend to verify the number of poems in Webb's corpus in which it is possible to establish a relationship between semantic content in terms of negative vs. positive sense—usually referred to with one word as "the sentiment"—and the sound produced by syllables in particular, stressed ones. We adopt a lexical approach, mainly using the database of 40 K entries made available by Brysbaert et al. 2014.

Thus, I will match the negative sentiment expressed by the words' sense with sad-sounding rhymes and poetic devices as a whole, and the opposite for positive sentiment by scoring and computing the ratios. I repeat here below the way in which I organized vowel and consonant sounds:

- Low, middle, high-front, high-back

Where I identify the two classes low and middle as promoting positive feelings, and the two high as inducing negative ones. As to the consonants, I organized the sounds into three main classes and two types:

- *Obstruents* (plosives, affricates), *continuants* (fricatives), *sonorants* (liquids, vibrants, approximants) plus the distinction into
- Voiced vs. unvoiced.

In this case, the ratios are computed dividing the sum of continuants and sonorants by the number of obstruents; and the second parameter will be the ratio obtained by dividing number of voiced by unvoiced. Whenever the value of the ratios is above 1, positive results are obtained; the contrary applies whenever values are below 1. In this way, counting results is immediate and very effective.

The Result section of the paper has a first rather lengthy subsection dedicated to the problem of rhyming structure which in the Sonnets constitutes the basic framework onto which all the subsequent reasoning is founded. Another subsection is dedicated to associating rhyming schemes with different themes as they have evolved in time. We dedicate a subsection to explaining the importance of the lexical approach in organizing the rules for the system SPARSAR, which derives the final vowel and consonant grids that allow us to make the first comparison. The lexical and semantic approach to deriving the sentiment of each sonnet operates a first subdivision of harmonic and disharmonic sonnets into negatively vs. positively marked sonnets. Measuring correlations reveals a constant contrasting attitude induced by the sound–sense agreement, which we interpret as an underlying hidden intention to produce some form of ironic mood in Shakespeare's sonnets.

Detecting irony requires a much deeper and accurate analysis of the semantic and the pragmatics of the sonnets. We proceed into two separate but conjoined ways: producing a gold standard of the sonnets and then manually annotating each sonnet using the highly sophisticated labeling system proposed by the Appraisal Theory Framework, ATF that we introduce briefly in Section 3.2.4. Matching the empirical approach and the automatic analysis confirms the overall underlying hypothesis: the sound–sense disharmony has a fundamental task, that of suggesting an underlying ironic attitude which is at the heart of all the sonnets. ATF makes available a more fine-grained approach which takes non-literal language into due account, thus improving on the previous method of sentiment-based analysis (see Martin et al. [9] and Toboada et al. [10]).

## 2. Materials and Methods

In this section, I will present the system SPARSAR and the pipeline of modules that allow it to carry out the complex analysis reported above.

### 2.1. SPARSAR—A System for Poetry Analysis and Reading

SPARSAR [11] produces a deep analysis of each poem at different levels: it works at the sentence level at first, then at the verse level and finally at the stanza level (see Figure 1 below). The structure of the system is organized as follows: the input text is

processed at first at a syntactic and semantic level and grammatical functions are evaluated. Then, the poem is translated into a phonetic form, preserving its visual structure and its subdivision into verses and stanzas. Phonetically translated words are associated with mean duration values taking into account position in the word and stress. At the end of the analysis of the poem, the system can measure the following parameters: mean verse length in terms of msec. and in number of feet. The latter is derived by a verse representation into metrical structure. Another important component of the analysis of rhythm is constituted by the algorithm that measures and evaluates rhyme schemes at the stanza level and then the overall rhyming structure at the poem level. In addition, the system has access to a restricted list of typical pragmatically marked phrases and expressions that are used to convey specific discourse function and speech acts, and need specialized intonational contours.

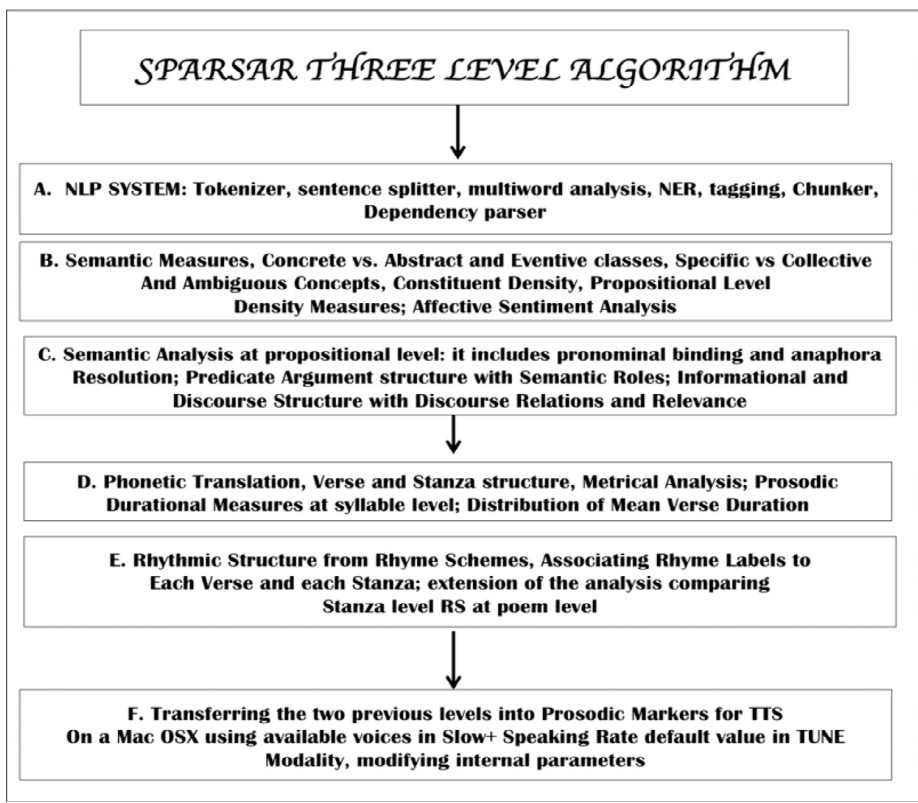

**Figure 1.** Architecture of *SPARSAR* with main pipeline organized into three levels.

We use the word "expressivity" [12], referring to the following levels of intervention of syntactic–semantic and pragmatic knowledge, which include the following:

- Syntactic heads which are quantified expressions;
- Syntactic heads which are preverbal subjects;
- Syntactic constituents that starts and ends an interrogative or an exclamative sentence;
- Distinguish realis from irrealis mood;
- Distinguish deontic modality including imperative, hortative, optative, deliberative, jussive, precative, prohibitive, propositive, volitive, desiderative, imprecative, directive and necessitative, etc.;
- Distinguish epistemic modality including assumptive, deductive, dubitative, alethic, inferential, speculative, etc.;
- Any sentence or phrase which is recognized as a formulaic or frozen expression with specific pragmatic content;
- Subordinate clauses with inverted linear order, distinguishing causal from hypotheticals and purpose complex sentences;
- Distinguishing parentheticals from appositives and unrestricted relatives;

- Discourse Structure to distinguish satellite and dependent clauses from the main clause;
- Discourse structure to check for discourse moves—up, down and parallel;
- Discourse relations to tell foreground relations from backgrounds;
- Topic structure to tell the introduction of a new topic or simply a change at relational level.

Current TTS only takes into account information coming from punctuation and, in some cases, from tagging. This hampers the possibility to capture the great majority of structures listed above. In addition, they do not adequately consider ambiguity of punctuation: for instance, the comma is a highly ambiguous punctuation mark with a whole set of different functions which are associated with specific intonational contours, and require semantic- and discourse-level knowledge to disentangle ambiguity. In general, punctuation marks like question and exclamative marks, are always used to modify the prosody of the previous word, which on the contrary is clearly insufficient to reproduce such pragmatically marked utterances and would encompass the whole sentence from its beginning word.

### 2.2. The Modules for Syntax and Semantics

The system uses a modified version of *VENSES*, a semantically oriented NLP pipeline [13]. It is accompanied by a module that works at sentence level and produces a whole set of analyses at quantitative, syntactic and semantic levels. As regards syntax, the system makes available chunks and dependency structures. Then, the system introduces semantics both in the version of a classifier and by isolating the verbal complex in order to verify propositional properties, like presence of negation, to compute factuality from a crosscheck with modality, aspectuality—that is derived from the lexica—and tense. On the other hand, the classifier has two different tasks: separating concrete from abstract nouns, and identifying highly ambiguous from singleton concepts (from number of possible meanings from WordNet and other similar repositories). Eventually, the system carries out a sentiment analysis of the poem, thus contributing a three-way classification: neutral, negative, and positive that can be used as a powerful tool for prosodically related purposes.

Semantics in our case not only refers to predicate–argument structure, negation scope, quantified structures, anaphora resolution and other similar items. It essentially refers to a propositional-level analysis, which is the basis for discourse structure and discourse semantics contained in discourse relations. It also paves the way for a deep sentiment or affective analysis of every utterance, which alone can take into account the various contributions that may come from syntactic structures like NPs and Aps, where affectively marked words may be contained. Their contribution needs to be computed in a strictly compositional manner with respect to the meaning associated with the main verb, where negation may be lexically expressed or simply lexically incorporated in the verb meaning itself.

In Figure 1 above the architecture of the deep system for semantic and pragmatic processing, in which phonetics are shown, prosodics and NLP are deeply interwoven. The system does low-level analyses before semantic modules are activated, that is tokenization, sentence splitting, and multiword creation from a large lexical database. Then, chunking and syntactic constituency parsing is conducted using a rule-based recursive transition network: the parser works in a cascaded recursive way to include higher syntactic structures up to the sentence and complex sentence level. These structures are then passed to the first semantic mapping algorithm that looks for subcategorization frames in the lexica freely made available for English, including a proprietor lexicon of some 10 K entries, with most frequent verbs, adjectives and nouns, also containing a detailed classification of all grammatical or function words. This mapping is performed following LFG principles [14,15], where c-structure is mapped onto f-structure, thus obeying uniqueness, completeness and coherence. The output of this mapping is a rich dependency structure, which contains information related to implicit arguments as well, i.e., subjects of infinitivals, participials

and gerundives. LFG representation also has a semantic role associated with each grammatical function, which is used to identify the syntactic head lemma uniquely in the sentence. When fully coherent and complete predicate argument structures have been built, pronominal binding and anaphora resolution algorithms are fired. The coreferential processed are activated at the semantic level. Discourse-level computation is conducted at the propositional level by building a vector of features associated with the main verb of each clause. They include information about tense, aspect, negation, adverbial modifiers, and modality. These features are then filtered through a set of rules which have the task to classify a proposition as either objective/subjective, factual/nonfactual, foreground/background. In addition, every lexical predicate is evaluated with respect to a class of discourse relations. Eventually, discourse structure is built, according to criteria of clause dependency where a clause can be classified either as coordinate or subordinate.

### 2.3. The Modules for Phonetic and Prosodic Analysis

The second set of modules is a rule-based system that converts graphemes of each poem into phonetic characters; it divides words into stressed/unstressed syllables and computes rhyming schemes at the line and stanza level. To this end, it uses grapheme-to-phoneme translations made available by different sources, amounting to some 500 K entries, and include the CMU dictionary (Freely downloadable from http://www.speech.cs.cmu.edu/cgi-bin/cmudict accessed on 6 July 2023), MRC Psycholinguistic Database (Freely downloadable from https://websites.psychology.uwa.edu.au/school/mrcdatabase/uwa_mrc.htm accessed on 6 July 2023), Celex Database [16], plus a proprietor database made of some 20,000 entries. Out-of-vocabulary words are computed by means of a prosodic parser implemented in a previous project [17], containing a big pronunciation dictionary which covers 170,000 entries, approximately. Besides the need to cover the majority of grapheme-to-phoneme conversions through the use of appropriate dictionaries, the remaining problems to be solved are related to ambiguous homographs like "import" (verb) and "import" (noun), and are treated on the basis of their lexical category derived from previous tagging. Eventually, there is always a certain number of out-of-vocabulary words (OOVW). The simplest case is constituted by differences in spelling determined by British vs. American pronunciation. This is taken care of by a dictionary of graphemic correspondences. However, whenever the word is not found, the system proceeds by morphological decomposition, splitting at first the word from its prefix and if that still does not work, its derivational suffix. As a last resource, an orthographically based version of the same dictionary is used to try and match the longest possible string in coincidence with the current OOVW. Then, the remaining portion of the word is dealt with by guessing its morphological nature, and if that fails, a grapheme-to-phoneme parser is used. Some of the OOVWs that have been reconstructed by means of the recovery strategy explained above are wayfarer, gangrened, krog, copperplate, splendor, filmy, seraphic, seraphine, and unstarred.

Other words we had to reconstruct are shrive, slipstream, fossicking, unplotted, corpuscle, thither, wraiths, etc. In some cases, the problem that made the system fail was the presence of a syllable which was not available in our database of syllable durations, *VESD* [17]. This problem has been coped with by manually inserting the missing syllable and by computing its duration from the component phonemes, or from the closest similar syllable available in the database. We only had to add 12 new syllables for a set of approximately 1000 poems that the system computed.

The system has no limitation on type of poetic and rhetoric devices; however, it is dependent on language: Italian line verse requires a certain number of beats and metric accents which are different from the ones contained in an English iambic pentameter. Rules implemented can demote or promote word-stress on a certain syllable depending on the selected language, line-level syllable length and contextual information. This includes knowledge about a word being part of a dependency structure either as dependent or as head.

As R. Tsur [18] comments in his introduction to his book, iambic pentameter has to be treated as an abstract pattern and no strict boundary can be established. The majority of famous English poets of the past, while using iambic pentameter, have introduced violations, which in some cases—as for Milton's Paradise Lost—constitute the majority of verse patterns. Instead, the prosodic nature of the English language needs to be addressed, at first. English is a stress-timed language as opposed to Spanish or Italian which are syllable-timed languages. As a consequence, what really matters in the evaluation of iambic pentameters is the existence of a certain number of beats—5 in normal cases, but also 4 in deviant ones. Unstressed syllables can number higher, as for instance in the case of exceptional feminine rhyme or double rhyme, which consists of a foot made of a stressed and an unstressed syllable (very common in Italian) ending the line—this is also used by Greene et al. [19] to loosen the strict iambic model. These variations are made to derive from elementary two-syllable feet, the iamb, the trochee, the spondee, and the pyrrich. According to the author, these variations are not casual, they are all motivated by the higher syntactic–semantic structure of the phrase. So, there can be variations as long as they are constrained by a meaningful phrase structure.

In our system, in order to allow for variations in the metrical structure of any line, we operate on the basis of syntactic dependency and have a stress demotion rule to decide whether to demote stress on the basis of contextual information. The rule states that word stress can be demoted in dependents in adjacency with their head in case they are monosyllabic words. In addition, we also have a promotion rule that promotes function words which require word stress. This applies typically to ambiguously tagged words, like "there", which can be used as an expletive pronoun in preverbal position, and be unstressed; but, it can also be used as locative adverb, in that case in postverbal position, and be stressed. For all these ambiguous cases, but also for homographs not homophones, tagging and syntactic information is paramount.

Our rule system tries to avoid stress clashes and prohibits sequences of three stressed/three unstressed syllables unless the line syntactic–semantic structure allow it to be interpreted otherwise. Generally speaking, prepositions and auxiliary verbs may be promoted; articles and pronouns never. An important feature of English vs. Italian is length of words in terms of syllables. As may be easily gathered, English words have a high percentage of one-syllable words when compared to Italian which, on the contrary, has a high percentage of 3/4-syllable words.

*2.4. Computing Metrical Structure and Rhyming Scheme*

Any poem can be characterized by its rhythm which is also revealing of the poet's peculiar style. In turn, the poem's rhythm is based mainly on two elements: meter, that is distribution of stressed and unstressed syllables in the verse, presence of rhyming and other poetic devices like alliteration, assonance, consonance, enjambments, etc., which contribute to poetic form at the stanza level. This level is combined then with syntax and semantics to produce the adequate breath groups and consequent subdivision: these will usually coincide with line-stop words, but they may continue to the following line by means of enjambments.

As discussed above, see Figure 1, the analysis starts by translating every poem into its phonetic form. After processing the whole poem on a line-by-line basis and having produced all phonemic transcription, the system looks for poetic devices. Here, assonances, consonances, alliterations and rhymes are analysed and then evaluated. Here, metrical structure is computed, that is the alternation of beats: this is performed by considering all function or grammatical words which are monosyllabic as unstressed. In particular, "0" is associated with all unstressed syllables, and a value of "1" to all stressed syllables, thus including both primary and secondary stressed syllables. Syllable building is a discovery process starting from longest possible phone sequences to shortest one. This is performed heuristically trying to match pseudo syllables with the syllable list. Matching may fail and will then result in a new syllable which has not been previously met. The assumption

is that any syllable inventory will be deficient, and will never be sufficient to cover the whole spectrum of syllables available in the English language. For this reason, a certain number of phonological rules has been introduced in order to account for any new syllable that may appear. Also, syntactic information is taken advantage of, which is computed separately to highlight chunks' heads as produced by the bottomup parser. In that case, stressed syllables take maximum duration values. Dependent words, on the contrary, are "demoted" and take minimum duration values.

Metrical structure is used to evaluate its distribution in the poem by means of statistical measures. As a final consideration, we discovered that even in the same poem, it is not always possible to find that all lines have an identical number of syllables, identical number of metrical feet and identical metrical verse structure. If we consider the sequence "01" as representing the typical iambic foot, and the iambic pentameter as the typical verse metre of English poetry, there is no poem strictly respecting it in our analyses. On the contrary, we found trochees, "10", dactyls, "100", anapests, "001"and spondees, "11". At the end of the computation, the system is used to measure two important indices: "mean verse length" and "mean verse length in no. of feet", that is, mean metrical structure.

Additional measures that we are able to produce are related to rhyming devices. Since we consider it important to take into account structural internal rhyming schemes and their persistence in the poem, the algorithm makes available additional data derived from two additional components: word repetition and rhyme repetition at the stanza level. Sometimes, "refrain" may also apply, that is, the repetition of an entire line of verse. Rhyming schemes together with metrical length are the strongest parameters to consider when assessing similarity between two poems.

Eventually, the internal structure of metrical devices used by the poet can be reconstructed: in some cases, stanza repetition at the poem level may also apply. To create the rhyming scheme, couples of rhyming lines are searched by trying a match recursively of each final phonetic word with the following ones, starting from the closest to the one that is further apart. Each time, both rhyming words and their distance are registered. In the following pass, the actual final line numbers are reconstructed and then an indexed list of couples, line number–rhyming line for all the lines is produced, including stanza boundaries. Eventually, alphabetic labels are assigned to each rhyming verse starting from A to Z. A simple alphabetic incremental mechanism updates the rhyme label. This may go beyond the limits of the alphabet itself and in that case, double letters are used.

*2.5. From Sentiment Analysis to the Deep Pragmatic Approach by ATF*

We based a first approach to detecting sound–sense harmony on sentiment analysis, which in our case encompasses both a lexical and a semantic analysis at the propositional level. More generally speaking, computational research on sentiment analysis has been based on the use of shallow features with a binary choice to train statistical model [20] that, when optimized for a particular task, will produce acceptable performance. However, generalizing the model to new texts is a hard task and, in addition, the sonnets contain a lot of nonliteral language. The other common approach used to detect irony, in the majority of the cases, is based on polarity detection. Sentiment analysis [21,22] is in fact an indiscriminate labeling of texts either on a lexicon basis or on a supervised feature basis, where in both cases, it is just a binary—ternary or graded—decision that has to be made. This is certainly not explanatory of the phenomenon and will not help in understanding what it is that causes humorous reactions to the reading of an ironic piece of text. It certainly is of no help in deciding which phrases, clauses or just multiwords or simply words, contribute to create the ironic meaning (see [23]).

Shakespeare's Sonnets are renowned for being full of ironic content [24,25] and for their ambiguity, thus sometimes reverting the overall interpretation of the sonnet. Lexical ambiguity, i.e., a word with several meanings, emanates from the way in which the author uses words that can be interpreted in more ways not only because they are inherently polysemous, but because sometimes the additional meaning they evoke can sometimes be

derived on the basis of the sound, i.e., homophone (see "eye" and "I" in sonnet 152). The sonnets are also full of metaphors which many times require contextualising the content to the historical Elizabethan life and society. Furthermore, there is an abundance of words related to specific language domains in the sonnets. For instance, there are words related to the language of economy, war, nature and to the discoveries of the modern age, and each of these words may be used as a metaphor of love. Many of the sonnets are organized around a conceptual contrast, an opposition that runs parallel and then diverges, sometimes with the use of the rhetorical figure of the chiasmus. It is just this contrast that generates irony, sometimes satire, sarcasm, and even parody. Irony may be considered in turn as what one means using language that normally signifies the opposite, typically for humorous or emphatic effect; and a state of affairs or an event that seems contrary to what one expects and is amusing as a result. As to sarcasm, this may be regarded the use of irony to mock or convey contempt. Parody is obtained by using the words or thoughts of a person but adapting them to a ridiculously inappropriate subject. There are several types of irony, though we select verbal irony which, in the strict sense, is saying the opposite of what you mean for outcome, and it depends on the extra-linguistic context [26]. As a result, satire and irony are slightly overlapping but constitute two separate techniques; eventually, sarcasm can be regarded as a specialization or a subset of irony. It is important to remark that in many cases, these linguistic structures may require the use of non-literal or figurative language, i.e., the use of metaphors.

Joining sentiment, irony and sound as they could have been heard by Elizabethan audiences is what makes the Sonnets so special even today, and our paper succeeds in clarifying the peculiarities of the at the same time deep and shallow combination of factors intertwined to produce the final glamorous result that every sonnet does also today.

## 3. Results

This section will present results of the analysis of Shakespeare's sonnets at first and then of Webb's poems highlighting all cases of harmony and disharmony with relation to theme and meaning intended in the poem.

### 3.1. Sound Harmony in the Sonnets

We postulate the existence of a hidden plan in Shakespeare's poetic approach, to abide to a harmonic principle that requires all varieties of sound classes to be present and to relate by virtue of a sound–meaning correspondence, to thematic and meaning development in the sonnet. To discover such a plan, we analysed the phonetic representation of the rhyming words of all sonnets using SPARSAR—the system that analyzes automatically any poem, see below—and then organized the results of all vowel sounds into the four classes mentioned above. We did the same with consonants and consonant clusters in order to obtain a sound grid that is complete and retains as much complexity as possible of each poem and compared it with sense-related analyses.

However, in order to produce such a result, almost 500 phonologically ambiguous rhyming words had to be checked and transformed into the pronunciation current in the XVIth century when Early Modern English was still existent. This will be explained in a section below. It is also important to remind that the sonnets contain some 800 contractions and some 50 metrical adjustments which require the addition of an end of word syllable. After all these corrections, we obtained a sound map which clearly testifies to the intention of preserving a sound–sense harmony in the overall poetic scheme of the sonnets.

We may state as a general principle that the sound–sense harmony is respected whenever there is a full agreement between the sound grid and the mood associated with the meaning of the words. We assume then that there exists a sound–meaning correspondence by which different emotions or sentiments may be associated with each class. And of course, different results will be obtained by subtracting one class from the set, as we will comment below.

### 3.1.1. Periods and Themes in the Sonnets

The sonnets have been written in the short period that goes from 1592 to 1608, but we do not know precisely when. The majority of critics have divided them up into two main subperiods: a first one from 1592 to 1597 encompassing Sonnets from 1 to 126 and a second subperiod from 1598 to 1608 that includes Sonnets 127 to 154 (see Melchiori [9]). In addition, the sonnets have been traditionally subdivided into five main cycles or themes (Melchiori: Introduction): from 1 to 17, the reproduction sonnets, progeny, in which the poet spurs the young man to marry; from 18 to 51, immortality of poetry, the temptation of the friend by the lady, friend is guilty, and the absence of the loved one; from 52 to 96, poetry and memory, beauty and poetic rivalry; from 97 to 126, memory, the mistakes of the poet; and the last one from 127 to 152, the theme of the dark lady and unfaithfulness.

In Michael Schoenfeldt's Introduction to his edited book [27], we find a similar subdivision: Sonnets 1–126 are addressed to a beautiful young man, while Sonnets 127–152 are directed to a dark lady, and there are many other thematic and narrative sequences like 1–17 mentioned above (ibid. iii).

In the study of inversion made by Ingham and Ingham [28] on all of Shakespeare's plays, the authors reported three separate historical periods characterized by different frequencies in the use of subject inversion (VS) compared with canonical order (SV) on a total number of 951 clause structures:

1. A first period that goes from 1592 to 1597, where we have the majority of the cases of VS (214 over 421 total cases).
2. A second period that goes from 1598 to 1603, where the number of cases is reduced by half, but the proportion remains the same (109 over 213 total cases). A third period that goes from 1604 to 1608, where the proportion of cases is reverted (95 over 317 total cases) and VS cases are the minority.

The main themes of the sonnets are well-known: from 1 to 126 they are stories about a handsome young man, or rival poet; from 127 to 152 the sonnets concern a mysterious "dark" lady the poet and his companion love. The last two poems are adaptations from classical Greek poems. In the first sequence, the poet tries to convince his companion to marry and have children who will ensure immortality. Aside from love, the poem and poetry will "defeat" death. In the second sequence, both the poet and his companion have become obsessed with the dark lady, the lexicon used is sensual and the tone distressing. These themes are at their highest in the best sonnets indicated above. So, we would expect these sonnets to exhibit properties related to popularity that set them apart from the rest.

We decided to look into the "themes" matter more deeply and discovered that the immortality theme is in fact present through the lexical field constituted by the keyword DEATH. We thus collected all words related to this main keyword and they are the following ones, omitting all derivations, i.e., plurals for nouns, third person, past tense and gerundive forms for verbs:

BURY, DEAD, DEATH, DECEASE, DECAY, DIE, DISGRACE, DOOM, ENTOMBED, GRAVE, GRIEF, GRIEV ANCE, GRIEVE, SCYTHE, SEPULCHRE, TOMB, and WASTE

Which we connected to SAD, SADNESS, UNHAPPYNESS, and WRINKLE. We ended up by counting 64 sonnets containing this lexical field which can be safely regarded as the most frequent theme of all. We then looked for the opposite meanings, the ones related to LIFE, HAPPY, HAPPYNESS, PLEASURE, PLEASE, MEMORY, POSTERITY, and ETERNITY. In this case, 28 sonnets are the ones mentioning these themes. So, overall, we individuated 92 sonnets addressing emotionally related strong themes. When we combine the two contrasting themes, death/eternity, sadness/memory, we come up with the following 19 sonnets:

1, 3, 6, 8, 15, 16, 25, 28, 32, 43, 48, 55, 63, 77, 81, 92, 97, 128, 147

### 3.1.2. Measuring All Vowel Classes

We show in the Table 1. below general statistics of the distribution of stressed vowel sounds in rhyming words of all the sonnets. We included also diphthongs, considering the

stressed portion as the relevant sound. The expected result is that the phonological class of high-back is the one less present in the sonnets, followed by high-front and low. Rhyming words with the middle stressed vowel are the ones with the highest frequency.

**Table 1.** Distribution of sounds of end-of-line rhyming words divided into four phonological classes.

| Phon. Class | High-Front | Mid | Low | High-Back | Total |
|---|---|---|---|---|---|
| No. Class | 119 | 159 | 142 | 111 | 531 |
| StrVowDiph | 493 | 861 | 587 | 314 | 2155 |

Here below are some examples of the classification of stressed vowels of rhyming words in the first three sonnets:

Sonnet 1: FRONT—increase, decease, spring, niggarding, be, thee;

> BACK—fuel, cruel;
> LOW—die, memory, eyes, lies;
> MIDDLE—ornament, content;

Sonnet 2: BACK—use, excuse, old, cold;

> MIDDLE—field, held, days, praise;
> LOW—lies,eyes, mine, thine, brow, now;

Sonnet 3: HIGH—thee, see, husbandry, posterity, be, thee;

> BACK—womb, tomb, viewest, renewest;
> LOW—another, mother, prime, time.

In Table 2, we show the presence of the four classes in each sonnet, confirming our starting hypothesis about the intention to maintain a sound harmony in each sonnet: as can be easily gathered, 140 sonnets over 154 have rhymes with sounds belonging to more than two classes.

**Table 2.** Subdivision of the sonnets by number of classes.

| No. Classes | 4-Class | 3-Class | 2-Class | 1-Class | Total |
|---|---|---|---|---|---|
| No. Sonnets | 77 | 64 | 12 | 1 | 154 |

There is one sonnet with only one class and it is sonnet 146; then, there are 13 sonnets with 2 classes of sounds: 8, 9, 64, 71, 79, 81, 87, 90, 92, 96, 124, and 149. These sonnets contain rhyming pairs with low and middle sounds, except for three sonnets: sonnet 71 which contains high-back and middle sounds; sonnet 9 which contains high-front and low sounds; and sonnet 96 containing high-front and middle sounds. The themes developed in these sonnets fit perfectly into the rhyming sound class chosen. Let us consider sonnet VIII which is all devoted to music and string instruments which require more than one string to produce their sound, thus suggesting the need to find a companion and get married. Consider the line "the true concord of well tunèd sounds," where hints to the need that sounds should be "well" tuned. Sonnet 81 celebrates the poet and his verse which shall survive when death will come. Sonnet 92 is in fact pessimistic in the possibility that love will last "for the term of life" and no betrayal will ensue. As to sonnet 146, it is a mixture of two seemingly different themes: a criticism of extravagant display or rich clothing of wealth by writers of the time, or perhaps his mistress and trying to convince her to change her ways for eternal salvation. Some critics regard this as the most profoundly religious or meditative sonnet. But, the feeling of the lover renouncing something brings back his mistress and the feeling of being powerless against her chastity, so that religious life becomes a desirable aim. In this sense, death can also be depicted as desirable.

It is important to notice the overall strategy of choice of sound in relation to meaning, in the rhyming devices used, for instance, in sonnet 147 (all sonnets are taken from the

online version made available at https://www.shakespeares-sonnets.com/ accessed on 6 July 2023):

> My reason, the physician to my love,
> Angry that his prescriptions are not kept,
> Hath left me, and i desperate now approve
> Desire is death, which physic did except.

The interesting fact in this case is that the appearance of a back high sound like |U| would match with the appearance of the saddest word, DEATH in the same stanza. In other words, the magistral use of rhyming sounds goes hand in hand with the themes and meaning developed in the sonnet.

Interesting to note how the rhyming sound evolves in the Sonnets taking sonnet 107 as an example: from SAD sounds (back and high), to MID and CLOSE to LOW and OPEN in the third stanza, to end with a repetition of MID sounds in the couplet:

> Not mine own fears, nor the prophetic soul
> Of the wide world dreaming on things to come,
> Can yet the lease of my true love control,
> Supposed as forfeit to a confin'd doom.
> The mortal moon hath her eclipse endur'd,
> And the sad augurs mock their own presage;
> Incertainties now crown themselves assur'd,
> And peace proclaims olives of endless age.
> Now with the drops of this most balmy time,
> My love looks fresh, and death to me subscribes,
> Since, spite of him, I will live in this poor rime,
> While he insults o'er dull and speechless tribes:
> And thou in this shalt find thy monument,
> When tyrants' crests and tombs of brass are spent.

In Sonnet 145, the overall feeling of sadness is transferred in the rhyming sounds: in the first stanza, the correct EME pronunciation requires |come| to be pronounced as |doom|, CUM/DUM a high-back sound which is then be repeated in the final couplet where "sav'd my life" appears. Here, important echoes of the |U| sound appear in the couplet with end-of-line words THREW and YOU.

> . . .. . .
> Straight in her heart did mercy come,
> Chiding that tongue that ever sweet
> Was us'd in giving gentle doom;
> . . .. . ..
> From heaven to hell is flown away.
> 'I hate', from hate away she threw,
> And sav'd my life, saying 'not you'.

We saw above the subdivision into classes; however, it does not tell us how the four phonological classes are distributed in the sonnets.

The resulting sound image coming from rhyme repetitions is eventually highlighted by the frequency of occurrence of same stressed vowel as shown in Table 3. In this table, we separated vowel sounds into three classes, high, middle, and low, to allow a better overall evaluation.

**Table 3.** Total count for vowel, final consonants and sonorant sounds organized into classes for all Shakespearean sonnets.

| N. | Un/Stress Vow/ Con | Following Vowel/ Consonant | Freq Occ | High | Middle | Low | Consonant |
|----|----|----|----|----|----|----|----|
| 1 | ay | d, er, f, l, m, n, r, t, v, z | 109 | | | 109 | |
| 2 | ey | d, jh, k, l, m, n, s, t, v, z | 81 | | 81 | | |
| 3 | n_ | d, iy, jh, s, t, z | 80 | | | | 80 |
| 4 | r_ | ay1, d, ey1, iy, iy1, k, n, ow, ow1, s, t, th, uw1, z | 68 | | | | 68 |
| 5 | eh | d, jh, k, l, n, r, s, t, th | 68 | | 68 | | |
| 6 | ih | d, l, m, n, ng, r, s, t, v | 51 | 51 | | | |
| 7 | ao | d, l, n, ng, r, s, t, th, z | 40 | | 40 | | |
| 8 | iy | d, f, ih, k, l, m, n, p, s, t, v, z | 45 | 45 | | | |
| 9 | s | iy, st, t | 38 | | | | 38 |
| 10 | uw | d, m, n, s, t, th, v, z | 47 | 47 | | | |
| 11 | ah | d, l, n, s, t, z | 34 | | | 34 | |
| 12 | ow | k, l, n, p, t, th, z | 21 | 25 | | | |
| 13 | t | er, ey1, iy, s, st | 21 | | | | 21 |
| 14 | ah | d, k, m, n, ng | 17 | | | 17 | |
| 15 | aa | n, r, t | 16 | | | 16 | |
| 16 | ae | ch, d, k, ng, s, v | 14 | | | 14 | |
| 17 | d_z | | 13 | | | | 13 |
| 18 | er | ay1, d, iy, z | 11 | | 11 | | |
| | | Total final sounds | 778 | 168 | 200 | 190 | 220 |

Eventually, we come up with 61 more frequent heads with occurrences up to four and a total of 778 repeated vowel and consonant line-ending sounds. We now consider the remaining 288 rhyming pairs organized into "head" and "dependent", i.e., the preceding end of the line's rhyming word and the one in the corresponding alternate/adjacent end of line.

A direct consequence of the level of rhyming pair repetition rate is the sound image created in each sonnet. We assume that a high level of repetition will create a sort of echo from one sonnet to the next and a continuation effect, but it will also contribute a sense of familiarity. We decided to verify what would be the overall sound effect created by the total number of rhyming pairs analysed. Thanks to SPARSAR modules for phonetic transcription and poetic devices detection discussed elsewhere [29], we managed to recover all correct rhyming pairs and their phonetic forms. We report the results in the tables below.

The resulting sound image coming from rhyme repetitions is eventually highlighted by the frequency of occurrence of same stressed vowel as shown in the two tables below. We separated vowel sounds into three classes, high, middle, and low, to allow for an easy overall evaluation. If we consider all vowel sounds, there appears to be a highly balanced use of rhyming pairs with stressed low vowels being the more frequent. Not so if we consider diphthongs—we always consider the stressed vowel in both rising and falling diphthongs.

### 3.1.3. Distributing Vowel and Diphthong Classes into Thematic Periods

Win Table 4 below, we collected all stressed vowels and diphthongs for the five periods or phases into which the Sonnets collection can be divided up and found interesting variations: Period 1 has only 17 sonnets and 238 stressed rhyming words; Period 2 has 34 sonnets and 476 rhyming words; Period 3 has the majority, 45 sonnets and 630 rhyming

words; Period 4 has 30 sonnets and 420 words; and Period 5 has the remaining 28 sonnets and 398 rhyming words.

**Table 4.** (**a**) Distribution of stressed rhyming vowels in five phases. (**b**) Weighted values of the distribution of stressed rhyming vowels in five Phases.

| (a) | | | | |
|---|---|---|---|---|
| | **Low** | **Middle** | **High** | **Total** |
| Period 1 | 40 | 42 | 57 | 139 |
| Period 2 | 105 | 68 | 102 | 275 |
| Period 3 | 111 | 105 | 136 | 352 |
| Period 4 | 59 | 79 | 122 | 260 |
| Period 5 | 66 | 60 | 99 | 225 |
| Totals | 381 | 354 | 516 | 1251 |
| (b) | | | | |
| | **Low** | **Middle** | **High** | **Total** |
| Period 1 | 2.3529 | 2.4706 | 3.3529 | 8.1765 |
| Period 2 | 3.0882 | 2 | 3 | 8.0882 |
| Period 3 | 2.4667 | 2.3334 | 3.0223 | 7.8223 |
| Period 4 | 1.9667 | 2.6334 | 4.0667 | 8.6667 |
| Period 5 | 2.3571 | 2.1429 | 3.5357 | 8.0357 |
| Totals | 30.529% | 28.365% | 41.106% | 100% |

In Table 4a we computed absolute values for each vowel class distributed in the five periods and what can be preliminarily noted is the high number of "high" vowels and the lower number of the two other classes. In Table 4b, we produced weighted measures in order to take into account differences in number of sonnets considered which, as a result, will produce a disparity in the total number of occurrences. Frequency values for each vowel class are now a ratio of the number of sonnets per phase, the same with total values.

In this case, we can easily see that high vowels are always the class which had the most occurrences and Periods 4 and 5 are the ones with the highest number—which, however, needs to be divided by two subclasses, front and back. The low vowel class is the one with higher percentage, and in Period 2, low vowels have their highest value when compared to the other Periods. The opposite takes place in Period 4, where High vowels are at their highest also compared with the other Periods and low vowels are at their lowest also compared to other Periods. We may note that, overall, the highest number of stressed vowels belongs to Phase 4, whereas the lowest number to Phase 3. Overall, the majority of stressed vowels belongs to the phonetic class of high vowels followed by low and then middle.

We must now consider diphthongs and verify whether the same picture applies. Diphthongs, as annotated in the CMU dictionary, do not contain any high stressed nuclear vowel, because the choice was to separate high vowels in all those cases. So, we are left with five diphthongs: two low, AW and AY; and three middle, EY, OW, and OY.

As can be easily gathered from absolute total values, middle diphthongs constitute by far the majority. In Table 5 below is their distribution in the five phases, and as we did in Table 4, we show at first absolute values and then in section (b) weighted values:

**Table 5.** (**a**) Distribution of stressed diphthongs in the sonnets divided in 5 phases. (**b**) Weighted valued of the distribution of stressed diphthongs in the sonnets in 5 phases.

| (a) | | | |
|---|---|---|---|
| | **Low** | **Middle** | **Total** |
| Phase 1 | 50 | 46 | 96 |
| Phase 2 | 78 | 103 | 181 |
| Phase 3 | 112 | 154 | 266 |
| Phase 4 | 81 | 72 | 153 |
| Phase 5 | 65 | 85 | 150 |
| Totals | 386 | 460 | 846 |
| (b) | | | |
| | **Low** | **Middle** | **Total** |
| Phase 1 | 2.9412 | 2.7059 | 5.6471 |
| Phase 2 | 2.2941 | 3.0294 | 5.3235 |
| Phase 3 | 2.4889 | 3.4223 | 5.9112 |
| Phase 4 | 2.7 | 2.4 | 5.1 |
| Phase 5 | 2.3214 | 3.357 | 5.3571 |
| Totals | 45.626% | 54.373% | 100% |

Both Phases 1 and 4 show a decrease of middle vs. low diphthongs, while the remaining three phases behave in the opposite manner: more middle than low diphthongs. The total distribution indicates Phase 3 as the highest number of diphthongs and Phase 4 as the lowest, just the opposite of the previous distribution. General totals show a distribution of middle vs. low diphthongs which is strongly in favour of middle ones. This is just the opposite of what we found in previous counts, and in part then compensates with the lack of high diphthongs.

Eventually, in Table 6 the overall sound image is determined by a strong presence of middle sounds, followed by low sounds and eventually high sounds.

**Table 6.** Sound image of the sonnets.

| | **Low** | **Middle** | **High** | **Total** |
|---|---|---|---|---|
| Vowels | 381 | 354 | 567 | 1312 |
| Diphthongs | 386 | 460 | | 854 |
| Total | 767 | 814 | 567 | 2166 |

*3.2. Rhyming and Rhythm: The Sonnets and Poetic Devices*

3.2.1. Contractions vs. Rhyme Schemes

Contractions are present in a great number in the sonnets. Computing them requires reconstructing their original complete corresponding word form in order to be able to match it to the lexicon or simply derive the lemma through morphological processing. This is essentially due to the fact that they are not predictable and must be analysed individually. Each type of contraction has a different manner to reconstruct the basis wordform. In order to understand and reconstruct it correctly, each contraction must go through recovering of the lemma. We have found 821 contractions in the collection, where 255 are cases of genitive's, and 167 are cases of past tense/participle 'd. The remaining cases are organised as follows:

- SUFFIXES attached at word end, for example ('s, 'd, 'n, 'st, 't, (putt'st));
- PREFIXES elided at word beginning, for example ('fore, 'gainst, 'tis, 'twixt, 'greeing);
- INFIXES made by consonant elision inside the word (o'er, ne'er, bett'ring, whate'er, sland'ring, whoe'er, o'ercharg'd, 'rous).

Now, consider a contracted word like "sland'ring": as said before, at first the complete wordform must be reconstructed in order to use it for recovering the lemma and using the grammatical category for syntax and semantics. However, when computing the metrical structure of each line, the phonetic translation should be made on the contracted word, which does not exist in any dictionary neither in the form "slandring" nor in the form "sland-ring". What we carry out is finding the phonetic transcription, if already existent, in the dictionaries, and then subtracting the phoneme that has been omitted, creating in this way a new word. This is okay until we come to the module where metrical counts are made on the basis of the number of syllables. But, here again, the phonetic form derived from the complete word is not easy to accommodate. There are two possible subdivisions of the phonetic form s_l_ae_n_d_r_ih_ng (in ARPAbet characters): syllable 1: s_l_ae_n_d_; syllable 2: r_ih_ng. Syllable 1 does not correspond to the subdivision for the complete word which would be s_l_ae_n_|d_eh_|r_ih_ng. Luckily, the syllable exists independently, but this only happens occasionally. In the majority of the cases, the new word form produces syllables which are inexistent and need to be created ad hoc.

### 3.2.2. Rhythm and Rhyme Violations

In poetry, in particular in the tradition of the sonnets in Elizabethan times, poetic devices play a fundamental role. Sonnets in their Elizabethan variety had a stringent architecture which required the reciter to organize the presentation according to logical structure in the stanza structure, on the one side introducing the main theme, expanding and developing the accompanying subthemes, exploring consequences, finding some remedies to solve the dilemma or save the protagonist. On the other side, the line-by-line structure required the reciter to respect the alternate rhyming patterns which were usually safeguarded by end-stopped lines. Thus, the audience expectations were strongly influenced by any variation related to rhyming and rhythm as represented by the sequence of breath groups and intonational groups. Whenever the rhyming pattern introduced a new unexpected pronunciation—not in other contexts—of a rhyming word, the audience was stunned: say a common word like *love* was pronounced to rhyme with *prove*. The same effect must have been produced with enjambments, whenever lines had to run-on because meaning required the syntactic structure to be reconstructed—as for instance, in lines ending in a head noun which had its prepositional-of modifier in the beginning of the following line. Breath groups and intonational groups had to be recast to suit the unexpected variation, but rhyming had to be preserved. We will explore these aspects of the sonnets thoroughly in this section.

In a previous paper [30], we discussed the problem of (pseudo) rhyme violations as it has been presented in the literature on Shakespeare. In particular, we referred to the presence of more than 100 apparent rhyme violations, that is, rhyming end-of-line words which according to current pronunciation do not allow the rhyming scheme of the stanza to succeed, but it did in the uncertain grammar of Early Modern English. For instance, in sonnet 1, we find two lines 2–4 with the stanza rhyme scheme ABAB, ending with the words die-memory. In this case, the second word *memory* should undergo a phonological transformation and be pronounced "memo'ry"(*memoray*) ending in a diphthong at the end and sounding like "die"/(*dye*). Linguist David Crystal has discussed and reported on this question in many papers and also on a website—http://originalpronunciation.com/ (accessed on 6 July 2023). He collects and comments rhyming words whose pronunciation is different from Modern RP English pronunciation, listing more than 130 such cases in the Sonnets. However, in our opinion, what is missing is a rigorous proposal to cope with the problem of rhyme violation, and the list of transformations contains many mistakes when compared with the full transcription of the sonnets published in [31]. The solution is lexical

as we showed in a number of papers [29,30], i.e., variations should be listed in a specific lexicon of violations and the choice determined by an algorithm. Here below is an excerpt of the table, where we indicated the number of the sonnet, the line number, the rhyming word pair, their normal phonetic transcription using ARPAbet and in the last column the adjustment provided by the lexicon as shown in the example reported here below.

Example of lexical treatment of rhyme violation.

| Sonnet No. | Line No. | Rhyme Violation | Arpabet Phoneme | Adjusted Phoneme |
|---|---|---|---|---|
| Sonnet 1 | 2–4 | die-memory | d_ay-m_eh1_m_er_iy | iy→ay |

Variants are computed by an algorithm that takes as input the rhyming word and its stressed vowel from the first line in a rhyming pair and compares it with the rhyming word and vowel of the alternate line. Here, as in the following pages, we will use the phonetic alphabet called ARPAbet which is the one of the phonetic dictionary made available by CMU for computational purposes. The phonetic annotation makes use of American English but includes all vowel phonemes of British English: it has 12 vowels, and two semiconsonants. The missing part regards diphthongs: there are eight diphthongs in the chart, but three of them—descending diphthongs—never appear in the CMU dictionary or are treated as a sequence of a semivowel and a stressed vowel—IA (for CLEAR, _ih_), EA (for DOWNSTAIR CAREFUL, eh_), and UA (for ACTUAL, w_ah). In case of failure, the lexicon of Elizabethan variants is searched. The same stressed vowel may undergo a number of different transformations, so it is the lexicon that drives the change, and it is impossible to establish phonological rules at feature level. Some words may be pronounced in two manners according to rhyming constraints; thus, it is the rhyming algorithm that will decide what to do with the lexicon of variants. The lexicon in our case has not been built manually but automatically, by taking into account all rhyming violations and transcribing the pair of words at line end on a file. The algorithm searches couples of words in alternate lines inside the same stanza and in sequence when in the couplet, and whenever the rhyme is not respected, it writes the pair in output. Take for instance the pair LOVE/PROVE, in that order in alternate lines within the same stanza: in this case, it is the first word that has to be pronounced like the second. The order is decided by the lexicon: LOVE is included in the lexicon with the rule for its transformation; PROVE is not. In some other cases, it is the second word that is modified by the first one, as in CRY/JOLLITY; again, the criterion for one vs. the other choice is determined by the lexicon.

In Table 7. below, we list the total number of violations we found subdividing them by five phases as we did before, in order to verify whether the conventions dictated by Early Modern English grammars of the time did eventually impose a standard in the last period, beginning with the XVIIth century. After Total, we indicate the total number of violations found followed by slash and the number of sonnets. The ratio gives a weighted number that can be used to compare different occurrences in the five phases. As can be noted, the highest number of violations are to be found in the first two phases. Then, there is a decrease from Phase II to Phase IV which is eventually followed by a slight increase in Phase V which, however, is lower than what we found in previous phases. The first two phases then have numbers well over the average: the decrease in the following phases testifies to a tendency in Shakespeare's work to fix pronunciation rules in the sonnets as more and more grammarians tried to document what constituted the rules for Early Modern English.

**Table 7.** Number of rhyme violations x five phases.

|  | Sonnets Interval | No. Rhyme Violations/ No. Sonnets | Ratio % |
|---|---|---|---|
| Phase I | 1–17 | 22/17 | 1.2941 |
| Phase II | 18–51 | 40/34 | 1.1765 |
| Phase III | 52–96 | 34/45 | 0.7556 |
| Phase IV | 97–126 | 18/30 | 0.6 |
| Phase V | 127–154 | 23/28 | 0.8214 |
| Total |  | 137/154 | 0.8896 |

We call these (pseudo) rhyming violations because current reciters available on Youtube do not dare use the old pronunciation required and produce a rhyming violation by using Modern English pronunciation. One of these reciters is the famous actor John Gilgoud, who when reading Sonnet 66, correctly pronounces DESERT with its original meaning, but then in Sonnet 116 produces three violations when rhyming pairs required transformations that were clearly mandatory in Early Modern English, and they are |love| to be pronounced with the vowel of |remove| in lines 2/4, |come| to be pronounced with the vowel of |doom| in lines 10/12, and |loved| to be pronounced with the vowel of |proved| in the couplet. How do we know that these words should be pronounced in that manner and not in the opposite way—say |remove| as |love|, |doom| as |come| and |proved| as |loved|, as is being asserted by Ben Crystal son of David? There are three criteria that determine the way in which words should rhyme: the first one is the rhyming constraints which were so stringent at the time owing to the fact that poetry was only recited and not read on books. Okay, then, there are rhyming constraints but how do they work, in which direction? The direction is determined by two factors: the first one is determined by universal phonological principles, as for instance the one the governs phonological variations of vowel sounds—in the vowel shift of verbs or nouns due to morphological changes—which systematically changed "low" and "mid" features into "high" features and not vice versa [32]. The other factor is simply lexical: i.e., not all words will be subject to a transformation in that period. As a result, some words had double pronunciation. This was extensively documented in books and articles published at the time and written by famous poets like Ben Jonson and a great number of grammarians of the XVI and XVII century. All this information is made available by the famous historical phonologist Wilhelm Vietor of the XIX century in a book published at first in 1889 (2 (we use 1909 Vol 2. edition that can be freely visualized at: https://books.google.it/books?id=rhEQAwAAQBAJ&printsec=frontcover&hl=it&source=gbs_ge_summary_r&cad=0#v=onepage&q&f=false accessed on 6 July 2023), by the title "A Shakespeare Phonology" which we have adopted as our reference. Variants are then lexically determined. Some words involved in the transformation are listed below using ARPAbet as the phonetic alphabet in the excerpt taken from the lexicon. As can be easily noticed, variants are related also to stress position, but also to consonant sounds.

Lexicon 1.

shks(despised,d_ih2_s_p_ay1_s_t,ay1,ay1)
shks(dignity,d_ih2_g_n_ah_t_iy1,iy1,ay1).
shks(gravity,g_r_ae2_v_ah_t_iy1,iy1,ay1).
shks(history,hh_ih2_s_t_er_iy1,iy1,ay1).
shks(injuries,ih2_n_jh_er_iy1_z,iy1,iy1).
shks(jealousy,jh_eh2_l_ah_s_iy1,iy1,ay1).
shks(jollity,jh_aa2_l_t_iy1,iy1,ay1).
shks(majesty,m_ae2_jh_ah_s_t_iy1,iy1,ay1).
shks(memory,m_eh2_m_er_iy1,iy1,ay1).

shks(nothing,n_ah1_t_ih_ng,ah1,ow1).

It is now clear that variants need to interact with information coming from the rhyming algorithm that alone can judge whether the given word, usually at line end—but the word can also be elsewhere—has to undergo the transformation or not. The lexicon in our case has not been built manually but automatically, by taking into account all rhyming violations and transcribing the pair of words at line end on a file. The algorithm searches couples of words in alternate lines inside the same stanza and whenever the rhyme is not respected, it writes the pair in output. Take for instance the pair LOVE/PROVE, in that order, in alternate lines within the same stanza: in this case, it is the first word that has to be pronounced like the second. The order is decided by the lexicon: LOVE is included in the lexicon with the rule for its transformation, PROVE is not. In some other cases, it is the second word that is modified by the first one, as in CRY/JOLLITY, again the criterion for one vs. the other choice is determined by the lexicon. Thus, the system SPARSAR has a lexicon of possible transformations which are checked by an algorithm that whenever a violation is found, it is searched for the word to be modified and alters the phonetic description. In case both words of the rhyming pair are in the lexicon, the type of variation to be selected is determined by the overall sound map of the sonnet: Shakespeare produced a careful sound harmony in the choice of rhyming pairs including four or at least three sound classes.

Commenting on David Crystal's Point of View

Since the rhyming scheme is a fundamental issue for establishing sound harmony, the problem constituted by rhyming violations needs a deeper inspection. David Crystal makes available on his website the full phonetic transcription of the sonnets. However, as said above, these transcriptions contain many mistakes. There are two vague explanations Crystal finds to support his transcriptions in his OP (Old Pronunciation) and the first is a tautology: the "pronunciation system has changed since the 16th century": this is what he calls "a phonological perspective" (ibid.:298). In Section 2, entitled "Phonological rhymes", he writes

"Far more plausible is to take on board a phonological perspective, recognizing that the reason for rhymes fail to work today is because the pronunciation system has changed since the 16th century. . . . a novel and illuminating auditory experience, and introduced audiences to rhymes and puns which modern English totally obscures. The same happens when the sonnets are rendered in OP. In sonnet 154, the vowel of "warmed" echoes that of "disarmed", "remedy" echoes "by", the final syllable of "perpetual" is stressed and rhymes with "thrall", and the vowel of "prove" is short and rhymes with "love".

And further on (ibid:299):

"Ben Jonson. . . wrote an "English Grammar" in which he gives details about how letters should be pronounced. How do we know that "prove" rhymed with "love"? This is what he says about letter "O" in Chapter 4: "It naturally soundeth. . .. In the short time more flat, and akind to "u;" as "cosen", "dosen", "mother", "brother", "love", "prove" ". And in another section, he brings together "love, glove" and "move". This is not to deny, of course, that other pronunciations existed at the time. . .. "Love" may actually have had a long vowel in some regional dialects, as suggested by John Hard (a Devonshire man) in 1570 (and think of the lengthening we sometimes hear from singers today, who croon "I lurve you"). But the overriding impression from the orthoepists is that the vowel in "love" was short. It is an important point, because this word alone affects the reading of 19 sonnets. . .."

The second one is the need to respect puns (ibid. 298) which work in OP but not in modern English and, finally, the idiosyncratic spellings in the First Folio and Quarto and the description of contemporary orthoepists, who often give real detail about how pronunciations were in those days. There are no phonological rules, not even a uniform criterion that underlies the variations. The first reason was expressed as follows at the beginning of the paper: "The pronunciation of certain words has changed between Early

Modern English and today, so that these lines (referring to sonnet 154 lines) would have rhymed in Shakespeare's time". The list of pronunciation variations in the Supplementary Material of his paper [33] is messy and confusing but what is more important is that it also contains many mistakes, and we will comment on the first 10 items below.

First of all, the new rhyming transformation of "loved" is not mentioned in the Supplementary Material where according to Crystal "a complete" list should have appeared (ibid.:299). But the most disturbing fact is the recital performed by Ben Crystal (his son and actor in the Globe Theater), which is courageously made publicly available on Youtube (at https://www.youtube.com/watch?v=gPlpphT7n9s accessed on 6 July 2023). We are given a reading of Sonnet 116 which is illuminating of the type of OP Crystal is talking about (see time point 6:12 of total 10:21). The reading in fact does not start there but further on in the last stanza. The first contradictory assertion is just here, in the first stanza where lines B should rhyme and LOVE should be made to rhyme with REMOVE (as it is suggested in the Supplementary Material). The question is that in sonnet 154, the same rhyming pair in the same order LOVE—>REMOVE is transcribed with the opposite pronunciation. In the same paper, he asserts that "the vowel of PROVE is short and rhymes with LOVE" (ibid.:298) referring to the couplet of Sonnet 154 which we assume should be also applied to the B rhyming pair in sonnet 116 and not give us lav/rimav, but rather luv/rimuv. Here, an important additional series of alliteration would be fired if we adopt this pronunciation which in fact is the rule all over the Sonnets: TRUE would rhyme with LOVE and REMOVE/R. But also, further on as we will see, LOVE will rhyme with FOOL and DOOM.

On p.296,
Let me not to the marriage of true minds
Admit impediments, love is not love
Which alters when it alteration finds,
Or bends with the remover to remove.

The recital starts in third stanza, continuing with the couplet.
Love's not Time's fool, though rosy lips and cheeks
Within his bending sickle's compass come;
Love alters not with his brief hours and weeks,
But bears it out even to the edge of doom:
If this be error and upon me proved,
I never writ, nor no man ever loved.

In the Supplementary Material, we find another mistake or contradiction, where Crystal wrongly transcribes "doom" to rhyme with "come" (came/dam) rather than the opposite (cum/dum) and "loved" to rhyme with "proved" (pravd/lavd) which again should be the opposite, (pruvd/luvd). Here, as elsewhere, for instance in Sonnet 55, DOOM rhymes with ROOM in the correct order, ROOM/DOOM, and with the correct sound. Again, let us consider Crystal's wrongly reporting in the Supplementary Material the rhyming pair LOVE/APPROVE as rhyming in the opposite manner, i.e., LOVE is being pronounced as APPROVE which is just the contrary in the transcription; APPROVE is being pronounced as LOVE with a short open-mid back sounds. In Crystal's words,

"There are 19 instances in the sonnets where "love" is made to rhyme with "prove", "move", and their derived forms. And when we look at the whole sequence, we find a remarkable 142 rhyme pairs that clash (13% of all lines). Moreover, these are found in 96 sonnets. In sum: only a third of the sonnets rhyme perfectly in modern English. And in 18 instances, it is the final couples which fails to work, leaving a particularly bad taste in the ear."

This is how he explains the list of the Supplementary Material:
. . .a complete list is given in the Supplementary Material to this paper. The list indicates a rhyming pair where the first element is the one to be transformed because otherwise violating the rhyme. For instance MEMORY = DIE (1) must be interpreted as

follows: pronounce "memory" with the same vowel of "die" in modern RP pronunciation to be found in sonnet 1.

It is important to note that the first element in most cases appears as the SECOND rhyming word in the pair, but in some other cases as the first word of the pair. But then, we find a long list of mistakes if we compare the expected pronunciation encoded in the Supplementary Material with the complete transcription of the sonnets made available by David Crystal in a pdf file in the same website, where results are turned upside down. For instance, LOVED = PROVED (116) has been implicitly turned into PROVED = LOVED, that is the transcription of the stressed vowel of "proved" is the same as the one of "loved" and not the opposite. More mistakes in the list can be found where words like TOMB and DOOM are wrongly listed in the opposite manner. In particular, DOOM is made to rhyme with the vowel of COME and not the oppositee; also, TOMB is made to rhyme with COME and DUMB reverting in both cases the order of the rhyming pair and of the transformation. The phonetic transcription file confirms the mistakes: in the related sonnets we find the same short mid-front vowel instead of a short U, dumb/tomb both in sonnet 83 and 101. In all of these cases, the head (the rhyming word of the first line) should be made to rhyme with the dependent (the rhyming word of the second line) as it happens in Sonnet 1 with MEMORY/DIE and in the great majority of cases. So, two elements must be taken into account: the order of the two words of the rhyming pair and then the commanding word, i.e., the word that governs the transformation. In the case of MEMORY/DIE, DIE is the head or the commanding word of the transformation, and comes first in the stanza, whereas MEMORY is the dependent word and comes as second line of the rhyming pair. We list below only the wrong cases and comment the type of mistake made, i.e., either as reverted order, the first element of the pair comes before and it should be read as second; reverted order, the first element is in fact the one deciding the type of vowel to be used; else the order is correct, but the pronunciation chosen is wrong. To comment on the wrong pronunciation required by the rhyme we sometimes use the pronunciation indicated by Vietor in his book, and the phonetic transcription of all the sonnets Crystal made in his pdf file.

There are more mistakes in the Supplementary Material, here are some of them:

| | | |
|---|---|---|
| anon/alone | 75 | -should be alone/anon (Vietor:70) both the order and the governor are wrong. It should be: pronounce ALONE as ANON with a short or long /o/ |
| are/care | 48 | -the order should be care/are, but then the mistake is ARE transcribed like CARE [kEUR :r] |
| are/care | 112, 147 | -the order is correct but the transcription is wrong as before |
| are/compare | 35 | -the order should be compare/are, transcription correct |
| are/prepare | 13 | -the order should be prepare/are, transcription wrong: ARE is pronounced like PREPARE [pEUR :r] |
| are/rare | 52 | -order correct and in transcription ARE is like RARE [rEUR :r]—but it should be the opposite. RARE should sound like ARE, rare/are even though the line with RARE comes first. |
| beloved/removed | 25 | -order correct, but the transcription is wrong: remove is transcribed with the vowel of beloved |
| brood/blood | 19 | the order should be blood/brood: the transcription is also wrong BROOD is transcribed like BLOOD. see Vietor:87, whilst [u] in blood, flood, good, wood s. seems to be the usual Elizabethan sound. |
| dear/there | 110 | correct order but the pronunciation of DEAR is transcribed wrongly as [di:r] while the one of THERE is [thEUR :re] |
| doom/come | 107,116,145 | correct order but the pronunciation should be governed by DOOM, a short or long [u](Vietor:86): transcription of DOOM is instead with the vowel of COME |

We solved the problem by creating a lexicon of phonetic transformations and an algorithm that looked at first for a match in the rhyming word pair positioned in alternate

lines if in stanza, and in a sequence if in couplet. In case there was no match, the algorithm looks up the second word in the lexicon, and then the first word and chooses the one that is present. In case both are present in the lexicon, the decision is taken according to the position of the rhyming pair in the sonnet with respect to previous rhymes.

### 3.2.3. Rhyming Constraints and Rhyme Repetition Rate

If on the one side we have rhyme-apparent violations using the EME pronunciation to suit the rhyme scheme of the sonnet, on the other side, the Sonnets show a high "Repetition Rate" as computed on the basis of rhyming words alone. Due to the requirements imposed by the Elizabethan sonnet rhyme scheme, violations are very frequent, but they are not sufficient to allow the poet with the needed quantity of rhyming words. For this reason, it can be surmised that Shakespeare was obliged to use a noticeable amount of identical rhyming word pairs. The level of rhyming repetition is in fact fairly high in the sonnets, if compared with other poets of the same period, as can be gathered from the tables below. This topic has not gone unnoticed, as for instance [34], which indicates repetition of rhyming words as occurring in a limited number of consecutive adjacent sonnets, but does not give an overall picture of the phenomenon. In fact, as will be clear from the data reported below, the level of rhyming repetition is fairly high and reaches 65% of all rhyming pairs. In [34], we also find an attempt at listing all sonnets violating rhyme schemes which according to him amount to 25. However, as can be easily noticed in the list reported in the Supplementary Material, the number of sonnets violating the rhyme scheme is much higher than that.

To enumerate rhyming repetitions, we collected all end-of-line words with their phonetic transcription and joined them in alternate or sequential order as required by the sonnet rhyme scheme 1–3, 2–4, 5–7, 6–8, 9–11, 10–12, 13–14—apart from sonnet 126 with only 12 lines and a scheme in couplets aabbccddeeff, and sonnet 99 with 15 lines. Seven rhyming pairs for a total of 1078, i.e., 154 sonnets multiplied by 14 equal 2156 divided by two—less one 2155. In the tables reported as an Supplementary Material—the tables related to Rhyming Pair Repetition Rate have only been presented in Torino [30] at the conference and have not been published elsewhere—we only consider at first pairs with a frequency occurrence higher than 4, and we group together singular and plural of the same noun, and third person present indicative, d/n past with base form for verbs. We list pairs considering first occurrence as the "head" and following line as the "dependent". Rhyme may be sometimes determined by rules for rhyme violations as is the case with "eye". We include under the same heading all morphologically viable word forms as long as word stress is preserved in the same location, as said above, including derivations. We decided to separate highly frequent rhyming heads in order to verify whether less frequent ones really matter in the sense of modifying the overall sound image of the sonnets. For that purpose, we produce a first sound map below, limited to higher frequency rhyming pairs and only in a separate count we consider less frequent ones, i.e., hapax, trislegomena and dislegomena.

In many cases, the same pair is repeated in inverted order as for instance "thee/me" and "me/thee", "heart/part" and "part/heart", "love/prove" and "prove/love" but also "love/move" and "love/remove" and "approve/love" and "love/approve", "moan/gone" and "foregone/moan", "alone/gone" and "gone/alone", "counterfeit/set" and "unset/counterfeit", "worth/forth" and "forth/worth", "elsewhere/near" and "near/there", etc. "Thee" is made to rhyme with "me", but also with "melancholy", "posterity", "see". "Eye/s" are made to rhyme with almost identical monosyllabic sounding words like "die", "lie", "cries", "lies", "spies"; but also with "alchemy", "gravity", "history", "majesty", and "remedy", which require the conversion of the last syllable into a diphthong /ay/ preceded by the current consonant. Most of the rhyming pairs evoke a semantic or symbolic relation which is asserted or suggested by the context in the surrounding lines of the stanza that contain them. Just consider the pairs listed above where relations are almost explicit. However, as remarked by [34], rhyme repetition inside the same sonnet may have a different

goal: linking lines at the beginning of the sonnet to lines at the end as is the case with sonnet 134 and the rhyme pair "free/me" which reappears in the couple in reversed order. Similar results are suggested by repetition of rhyme pair "heart/part" in sonnet 46.

In Table 8. we did the same count with two other famous poets writing poetry in the same century, Sir Philip Sydney and Edmund Spenser. We wanted to verify whether the high level of rhyming pairs repetition might also apply to other poets writing love sonnets. The results show some remarkable differences in the degree of repetitivity. In Table 9, repeated rhyming pairs are compared to unique ones or hapax rhyming pairs in three Elizabethan poets. Percentages reported are a ratio of all occurrences of rhyming pairs. In the first column, types are considered and Sydney overruns Shakespeare and Spenser. When we come to Token repeating rate—i.e., counting all occurrences of each type and summing them up, we still have the same picture. Eventually, unique or unrepeated rhyming pairs are higher in Spenser than in Shakespeare and Sydney.

**Table 8.** Rhyme repetition rates in three Elizabethan poets.

| Author/ Quanti- Ties | Rhyme- Pair Repeat Types | Rhyme- Pair Repeat Token | Hapax or Unique Rhyme- Pairs |
|---|---|---|---|
| Shakespeare | 18.02% | 65.21% | 34.79% |
| Spenser | 17.84% | 47.45% | 53.55% |
| Sydney | 22.37% | 72.08% | 27.02% |

**Table 9.** Rhyme repetition word class-frequency distribution for Shakespeare's sonnets.

| X Typ | FX Tok | Sum FX | Sum FX + X | % Sum FX + X |
|---|---|---|---|---|
| 28 | 1 | 28 | 28 | 2.72 |
| 17 | 1 | 17 | 45 | 4.37 |
| 14 | 2 | 28 | 73 | 7.09 |
| 12 | 2 | 24 | 97 | 9.43 |
| 10 | 1 | 10 | 107 | 10.4 |
| 9 | 5 | 45 | 152 | 14.77 |
| 8 | 3 | 24 | 176 | 17.1 |
| 7 | 1 | 7 | 183 | 17.78 |
| 6 | 6 | 36 | 219 | 21.28 |
| 5 | 10 | 50 | 269 | 26.14 |
| 4 | 29 | 116 | 385 | 37.41 |
| 3 | 37 | 111 | 496 | 48.2 |
| 2 | 87 | 174 | 670 | 65.11 |
| 1 | 359 | 359 | 1029 | 100.0 |

Now, let us consider the distribution of rhyming words into the corpus of the sonnets. As to general frequency data, the Sonnets contain a number of tokens equal to 18,283 with 3085 types, so-called Vocabulary Richness that is used to measure the ability of a writer to use different words in a corpus, corresponds to 16.87%, a high value for that time when compared with other poets. Also, the number of Hapax and Rare Words (indicating the union of Hapax, Dis and TrisLegomena) corresponds to average values for other poets, respectively to 56%, the first type, and 79%, the second one. If we look at similar data for

rhyming words, we see that Rare Words cover more than 65% of all as can be gathered from Table 10 below:

**Table 10.** Quantitative data for six appraisal classes for sonnets with highest contrast.

|         | Appr.Pos | Appr.Neg | Affct.Pos | Affct.Neg | Judgm.Pos | Judgm.Neg |
|---------|----------|----------|-----------|-----------|-----------|-----------|
| Sum     | 56       | 25       | 53        | 77        | 32        | 122       |
| Mean    | 2.533    | 1.133    | 2.4       | 3.466     | 1.444     | 5.466     |
| St.Dev. | 8.199    | 3.691    | 7.732     | 11.202    | 4.721     | 17.611    |

We report for each word frequency type in column 1—there is only one head word (*thee*) with frequency 28—the corresponding number of tokens in Table 9, followed by the sum of tokens, the incremental sum and the corresponding percentage with respect to total corpus. As can be noticed from the last column, where incremental percent of rhyme-pair words corpus coverage is reported, the total of rare words, i.e., type rhyme-pair with frequency of occurrence lower than 4, is 62.59%, a fairly low value if compared to the measure evaluated on simple type/token ratios. If we look at most important English poets, as documented in a previous paper , we can see that the average value for Rare Words is 77.88%. However, we are here dealing with rhyming words and the comparison may not be so relevant.

### 3.2.4. The Sound–Sense Harmony Visualized in Charts

As will appear clearly from the charts below, all the data show a contrasting behaviour which will be attested by correlation values. Where sentiment values increase, the corresponding values for vowels and consonants decrease. To allow better perusing of the trends we split the sonnets into separate tables according to whether their sentiment values are positive or negative. The first chart contains the eleven sonnets which received the highest positive sentiment values. All the charts are drawn from the tables of data derived from the analysis files in xml format, which will be made available as supplementary data (please see Figure 2).

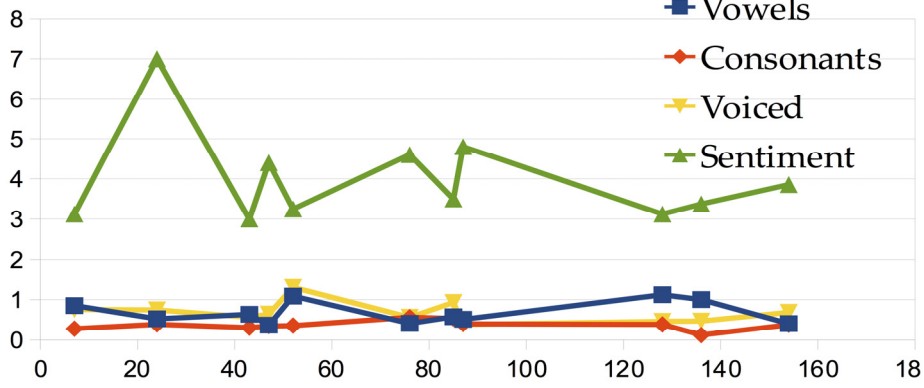

**Figure 2.** The eleven most positively marked sonnets: 7, 24, 43, 47, 52, 76, 85, 87, 128, 136, 154.

As can be easily noticed, all sound data seem to agree, showing a trend which is very close for the three variables. On the contrary, the sentiment variable has strong peaks and its values are set apart from the sound values. However, the interval of variability for sound variables does remain below or close to 1, thus indicating an opposite trend. In particular, consonants are all below 1, vowels oscillate in three cases, 52, 128, and 136, voiced in two cases, 52 and 85, in this case still below 1 but very close 93% in favour of unvoiced.

We interpret consistently contrasting values as a way to convey ironic, sarcastic and sometimes parodistic meaning. More on this interpretation below. Sonnet 136 is the one that is highly ambiguous and consequently ironic, celebrating the "Will" or simply "will".

Sonnet 128 is all devoted to music and playing with a wooden instrument which is the target of the ironic vein and the double meaning of words like "tickle". Finally, sonnet 52 is the celebration of the beloved as a "chest" where the rich keep their treasure, and which must be enjoyed "seldom". Sonnet 85 is a celebration of silent thought, and for this theme, it is filled with consonants which are continuants |h,f,th| and are unvoiced, but many words are marked by a sonorant syllable, thus voiced.

We now separate 16 sonnets which have sentiment equal to 1 or slightly lower than 1 but always higher than 92% in favour of positively marked. They are the following: 22, 33, 51, 60, 64, 73, 94, 97, 101, 102, 109, 118, 123, 131, 141, and 150 (please see Figure 3).

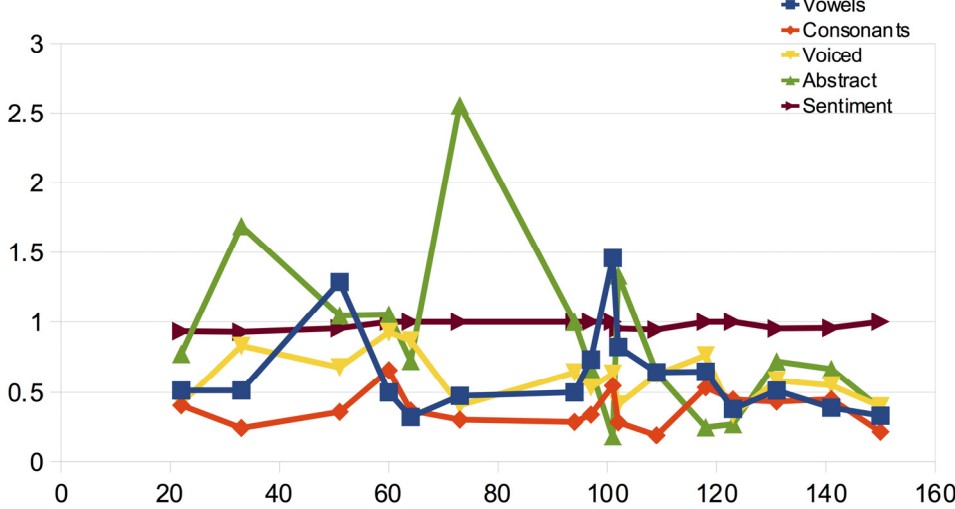

**Figure 3.** Chart of the 16 borderline sonnets positively marked for sentiment.

In this chart we added the ratio for Abstract/Concrete, which shows a peak for sonnet 73. As the chart clearly shows, the line for Sentiment borders 1, as to the remaining variables, Vowels is the one oscillating most after Abstract. Voiced and Consonants are fairly always aligned apart from sonnet 33 and 102. In both sonnets, the number of "Obstruents" (|b,d,p,t,k,g|) is very low and real consonants are substituted by "Continuants" (|s,sh,th,f,v,h|) both voiced and unvoiced. In the following analysis, for this reason, I will only consider Voicing as the relevant variable for consonants and this will show better agreement in the overall data. Now, we show charts for all negatively marked sonnets using only three variables, starting from Figure 4 below.

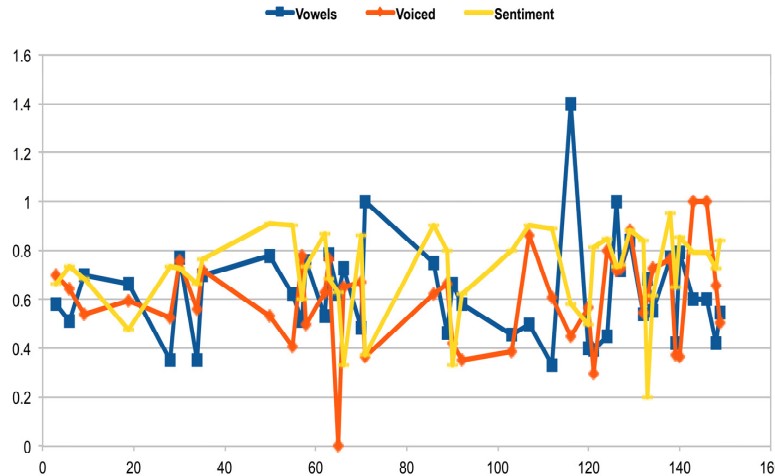

**Figure 4.** Chart of the 42 negatively marked sonnets: 3, 8, 9, 19, 28, 30, 34, 35, 50, 55, 57, 58, 60, 62, 63, 65, 66, 71, 86, 89, 92, 103, 107, 112, 116, 120, 121, 124, 126, 127, 129, 132, 133, 134, 138, 139, 140, 143, 146, 148, 149.

As can be easily seen, the Sentiment variable is always below 1 but the two remaining variables oscillate up and down, the vowel one oscillating most in the upper portion of the chart, and the voiced one in the lowe portion. In this case, the contrast is even stronger and correlations show a negative trend between Vowels and Sentiment: the one has a decreasing trend while the other has it increasing, apart from a few exceptions, sonnets 30, 35 and 127, which have almost identical values for the three variables. The other correlation between Voicing and Sentiment is positive but very weak: 0.1769.

Correlation between Vowel and Sentiment is positive but very weak; correlation between the Voicing parameter and Sentiment is again negative and very weak at $-0.0065037$. Thus, results for the 42 sonnets negatively marked by sentiment show that we have negative correlation between vowels and voicing, and vowels and sentiment, but positive correlation between voicing and sentiment. So, it is just the opposite of what we obtain with positively marked sonnets. And finally, in Figure 5. we show the eleven most positively marked sonnets show the same contrasting results.

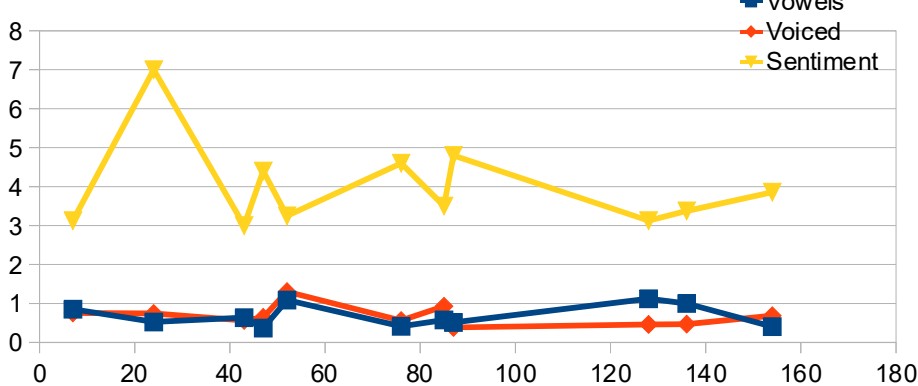

**Figure 5.** The eleven most positively marked sonnets show the same slightly positive correlation for Vowels–Voicing but very strong negative correlation between Vowels–Sentiment and slightly negative for Voicing/Sentiment at $-0.11423482$—colours in this case have no meaning.

As to the remaining 85 sonnets positively marked for sentiment, they all have very weak but positive correlations between sound and sense, i.e., below 0.1, respectively, 0.0387 for vowels, 0.05 for consonants, and 0.091 for voicing. The conclusion we may draw is that the sound–sense harmony in Shakespeare's sonnets is represented by a weak extended harmony for those positively marked for sentiment but a strong disharmony for those sonnets negatively marked for sentiment: in particular in all the sonnets we have an inverse correlation, between the two most important variables, Voicing (whether a consonant is a real Obstruent or not) and Sentiment. As said above, voicing includes real obstruents and unvoiced continuants: |p,t,k,s,sh,f,th|. When the pair Voicing/Sentiment assumes a positive correlation value, the other pair Vowel/Sentiment shows the opposite and is negative. Then, we saw the exceptions, in those sonnets which are most positively marked for sentiment, the correlations between Vowel and Sentiment are positive but the correlation between Voicing and Sentiment is negative. Sonnets negatively marked for sentiment have a positive correlation between Voicing and Sentiment, but a negative correlation between Vowel and Sentiment. In other words, the behaviour is just reversed: when meaning is positively marked the sound harmony verges towards a negative feeling. On the contrary, when the meaning is negatively marked the sound harmony verges, bends towards a positive sound harmony. I assume what Shakespeare intended to produce in this way was a cognitive picture of ironic poetic creation.

### 3.2.5. From Sentiment to Deep Semantic and Pragmatic Analysis with ATF

The final part of the analysis takes us deep into the hidden meaning that the sonnets communicate, i.e., irony. To carry that out, we need to substitute sentiment analysis with a much more semantically consistent framework that could allow us to enter the more

complex system of relational meanings that are governed by pragmatics. In this case, neither a word-by-word analysis or propositional-level analysis would be sufficient. We need to capture sequences of words which may have a non-literal meaning and associate appropriate labels: this is what the Appraisal Theory Framework can be useful for.

We have devised a sequence of steps in order to confirm experimentally our intuitions. The preliminary results obtained using sentiment analysis cannot be regarded as fully satisfactory for the simple reason that both the lexical and the semantic approach based on predicate-argument structures are unable to cope with the use of non-literal language. Poetic language is not only ambiguous but it contains metaphors which require abandoning the usual compositional operations for a more complex restructuring sequence of steps.

This has been carefully taken into account when annotating the sonnets by means of Appraisal Theory Framework (henceforth ATF). In our approach, we have followed the so-called incongruity presumption or incongruity-resolution presumption. Theories connected to the incongruity presumption are mostly cognitive-based and related to concepts highlighted, for instance, in [35]. The focus of theorization under this presumption is that in humorous texts, or broadly speaking in any humorous situation, there is an opposition between two alternative dimensions. As a result, we have been looking for contrast in our study of the sonnets, produced by the contents of manual classification. Thus, we have used the Appraisal Framework Theory [36]—which can be regarded as the most scientifically viable linguistic theory for this task, as has already been conducted in the past by other authors (see [12,37] but also [38]), showing its usefulness for detecting irony, considering its ambiguity and its elusive traits.

Thus, we proceeded like this: we produced a gold standard containing strong hints in its classification in terms of humour, by collecting most important literary critics' reviews of the 154 sonnets (the gold standard will be made available as Supplementary Material). To show how the classification has been organized we report here below two examples:

- SONNET 8

    SEQUENCE: 1–17 Procreation MAIN THEME: One against many ACTION: Young man urged to reproduce METAPHOR: Through progeny the young man will not be alone NEG.EVAL: The young man seems to be disinterested POS.EVAL: Young man positive aesthetic evaluation CONTRAST: Between one and many

- SONNET 21

    SEQUENCE: 18–86 Time and Immortality MAIN THEME: Love ACTION: The Young man must understand the sincerity of poet's love METAPHOR: True love is sincere NEG.EVAL: The young man listens the false praise made by others POS.EVAL: Young Man positive aesthetic evaluation CONTRAST: Between true and fictitious love.

As can be seen, the classification is organized using seven different linguistic components: we indicate SEQUENCE for the thematic sequence into which the sonnet is included; this is followed by MAIN THEME which is the theme the sonnet deals with; ACTION reports the possible action proposed by the poet to the protagonist of the poem; METAPHOR is the main metaphor introduced in the poem sometimes using words from a specialized domain; NEG.EVAL and POS.EVAL stand for Negative Evaluation and Positive Evaluation contained in the poem in relation to the theme and the protagonist(s); finally, CONTRAST is the key to signal presence of opposing concrete or abstract concepts used by Shakespeare to reinforce the arguments purported in the poem. Not all the sonnets were amenable to a pragmatic/linguistic classification. We ended up with 98 sonnets classified over 154, corresponding to a percentage of 63.64%, the rest have been classified as Blank. Many sonnets have received more than one possible pragmatic category. This is due to the difficulty in choosing one category over another. In particular, it has been particularly hard to distinguish irony from satire, and irony from sarcasm. Overall, we ended up with 54 sonnets receiving a double marking over 98. This was also one of the reasons to use ATF: often literary critics were simply hinting at "irony" or "satire", but the annotation gave us a

precise measure of the level of contrast present in each of the sonnets regarded generically as "ironic".

The annotation has been organized around only one category, *Attitude*, and its direct subcategories, in order to keep the annotation at a more workable level, and to optimize time and space in the XML annotation. *Attitude* includes different options for expressing positive or negative evaluation, and expresses the author's feelings. The main category is divided into three primary fields with their relative positive or negative polarity, namely:

- *Affect* is every emotional evaluation of things, processes or states of affairs, (e.g., like/dislike); it describes proper feelings and any emotional reaction within the text aimed towards human behaviour/process and phenomena.
- *Judgement* is any kind of ethical evaluation of human behaviour, (e.g., good/bad), and considers the ethical evaluation on people and their behaviours.
- *Appreciation* is every aesthetic or functional evaluation of things, processes and state of affairs (e.g., beautiful/ugly; useful/useless), and represent any aesthetic evaluation of things, both man-made and natural phenomena.

Eventually, we ended up with six different classes: *Affect Positive*, *Affect Negative*, *Judgement Positive*, *Judgement Negative*, *Appreciation Positive*, and *Appreciation Negative*. Overall, in the annotation, there is a total majority of positive polarities with a ratio of 0.511, in comparison to negative annotations with a ratio of 0.488. In short, the whole of the positive poles is 607, and the totality of the negative poles is 579 for a total number of 1186 annotations. *Judgement* is the more interesting category because it allows social moral sanction, which is then split into two subfields, *Social Esteem* and *Social Sanction*—which, however, we decided not to mark. In particular, whereas the positive polarity annotation of *Judgement* extends to *Admiration* and *Praise*, the negative polarity annotation deals with *Criticism* and *Condemnation* or *Social Esteem* and *Social Sanction* (see [38], p. 52). Here below is the list of 77 sonnets manually classified with ATF over 98 matching critics' evaluation.

**List of 77 successfully classified sonnets matching critics' evaluation.**

> **1 2 4 5 6 10 12 14 17 18 19 20 21 27 30 32 33 34 35 37 41 42 47 48 50 56 57 61 65 67 68 69 71 72 74 75 77 78 79 81 82 84 87 92 95 97 98 101 102 104 106 108 109 111 113 114 115 116 123 125 126 127 129 134 136 137 139 142 144 145 146 149 151 152 153 154**

As a first result, we may notice a very high convergence existing between critics' opinions and the output of manual annotation by *Appraisal* classes: 77 over 98 corresponds to a percentage of 78%. As to the sonnets' structure, *Judgement* is found mainly in the final couplet of the sonnets (for more details, see [3]). As to interpretation criteria, we assumed that the sonnets with the highest contrast could belong to the category of **Sarcasm**. The reason for this is justified by the fact that a high level of *Negative Judgements* accompanied by *Positive Appreciations* or *Affect* is by itself interpretable as the intention to provoke a sarcastic mood. As a final result, there are 44 sonnets that present the highest contrast and are specifically classified according to the six classes above. There is also a group that contains ambiguity sonnets which have been classified with a double class, mainly by **Irony** and **Sarcasm**. As a first remark, in all these sonnets, negative polarity is higher than positive polarity with the exception of sonnet 106. In other words, if we consider this annotation as the one containing the highest levels of *Judgement*, we come to the conclusion that a possible **Sarcasm** reading is mostly associated with presence of *Judgement Negative* and in general with high *Negative* polarity annotations. In Figure 6 below, we show the 44 sonnets classified with Sarcasm.

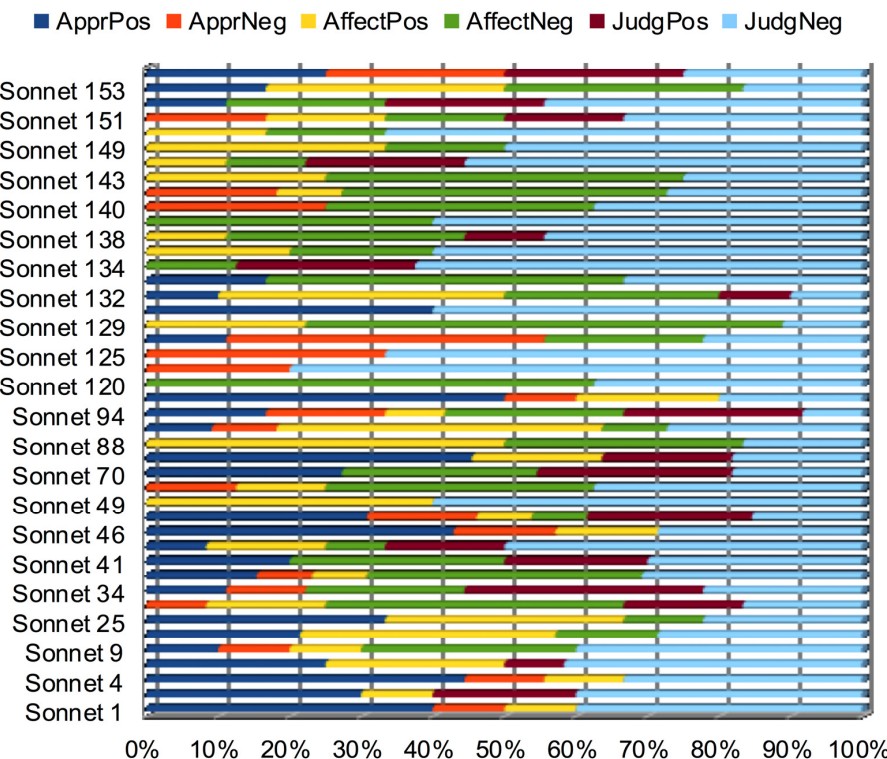

**Figure 6.** The 44 sonnets classified with Sarcasm with the highest level of Judgements—colours in this case have no meaning.

We associated different colours to make the subdivision into the six classes visually clear. It is possible to note the high number of *Judgements* both *Negative* (in orange) and *Positive* (in pale blue): in case *Judgement Positive* is missing, it is substituted by *Affect Positive* (pale green) or by *Appreciation Positive* (blue). This applies to all 44 sonnets apart from sonnets 120 and 121 where *Judgement Negative* is associated with *Affect Negative* and to *Appreciation Negative*. In other words, if we consider this annotation as the one containing the highest levels of *Judgement*, we come to the conclusion that possible **Sarcasm** reading is mostly associated with presence of *Judgement Negative* and, in general, with high *Negative* polarity annotations. As a first result, we may notice a very high correlation existing between critics' opinions as classified by us with the label highest contrast and the output of manual annotation by *Appraisal* classes.

In Figure 7 we show the group of 50 sonnets classified, mainly or exclusively, with **Irony** and check their compliance with *Appraisal* classes.

As can be easily noticed, the presence of *Judgement Negative* is much lower than in the previous diagram for **Sarcasm**. In fact, in only half of them—25—have annotations for that class; the remaining half introduces two other negative classes: mainly *Affect Negative*, but also *Appreciation Negative*. As to the main *Positive* class, we can see that it is no longer *Judgement Positive*, but *Affect Positive* which is present in 33 sonnets (please see Table 11).

**Table 11.** Quantitative data for six appraisal classes for sonnets with lowest contrast.

|  | Appr.Neg | Appr.Pos | Affct.Pos | Affct.Neg | Judgm.Pos | Judgm.Neg |
|---|---|---|---|---|---|---|
| Sum | 139 | 65 | 64 | 81 | 59 | 37 |
| Mean | 5.346 | 2.5 | 2.461 | 3.115 | 2.269 | 1.423 |
| St.Dev. | 18.82 | 8.843 | 8.707 | 11.009 | 8.029 | 5.047 |

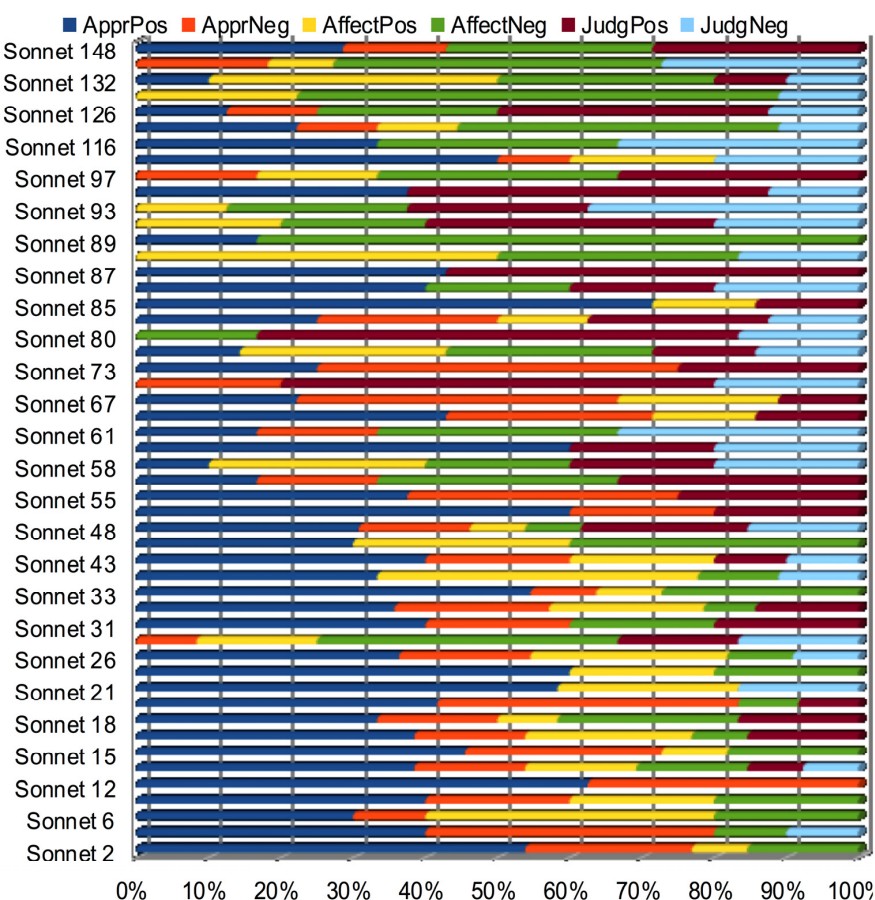

**Figure 7.** The 50 sonnets classified with Irony, with a lower level of Judgement Negative but higher Affect Negative.

In other words, we can now consider that **Sarcasm** is characterized by a majority of negative evaluations 146/224 while **Irony** is characterized by a majority of *Positive* evaluations 262/183 and that the values are sparse and unequally distributed.

The final figure, Figure 8, concerns the number of sonnets with blank or neutral evaluation by critics which amount to 60. As a rule, this group of sonnets should look different from the two groups we already analysed.

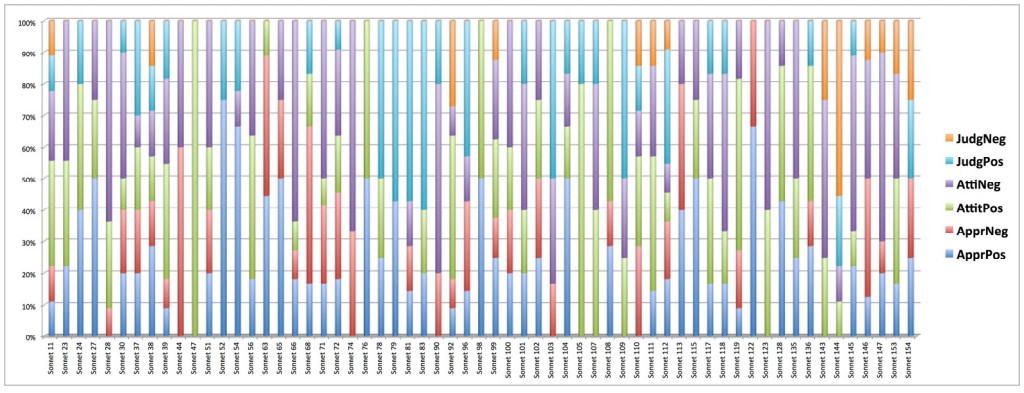

**Figure 8.** The 60 Sonnets classified by critics as neutral.

As expected, this figure looks fairly different from the previous two. The prevailing colour is pale blue, i.e., *Judgement Positive*; orange, i.e., *Appraisal Negative*, is only occasionally present; and green is perhaps the second prominent colour, i.e., *Affect Positive*. In order to

know how much the difference is, we can judge it from the quantities shown in Table 12 below.

**Table 12.** Quantitative data for six appraisal classes for sonnets with no contrast.

|         | Appr.Pos | Appr.Neg | Affct.Pos | Affct.Neg | Judgm.Pos | Judgm.Neg |
| ------- | -------- | -------- | --------- | --------- | --------- | --------- |
| Sum     | 88       | 59       | 89        | 109       | 49        | 8         |
| Mean    | 3.034    | 2.034    | 3.068     | 3.758     | 1.689     | 0.275     |
| St.Dev. | 1.268    | 7.638    | 11.482    | 14.052    | 6.368     | 1.079     |

3.2.6. Matching ATF Classes with the Algorithm for Sound–Sense Harmony (ASSH)

The experiment with ATF classes matching critics' evaluation has been fairly successful, but how do these classes gauge with the Sound–Sense harmony? In order to check this, we transferred the data related to vowels and consonants and matched them with ratios of the three main ATF categories: *Appreciation Positive/Negative, Affect Positive/Negative*, and *Judgement Positive/Negative*. As in previous computation, all data below 1 will be interpreted as a case of superior *Negative Polarity* and the opposite when data are above 1. To allow a better view of the overall data, we split them into sonnets with contrast to the first group that we show in Figure 9, and sonnets with no contrast to the second group, that we show in Figure 10. This time, however, we used our classification and abandoned the critics' one.

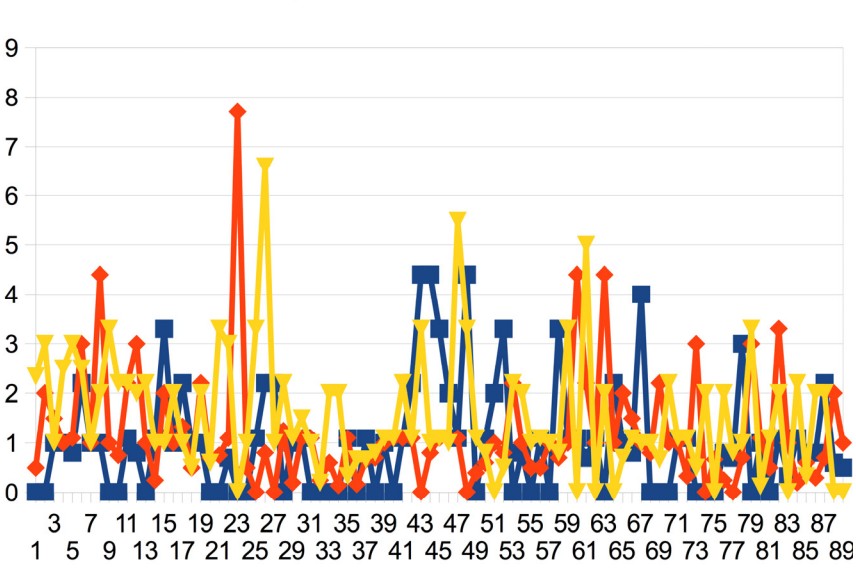

**Figure 9.** Distribution of 89 sonnets manually classified by ATF with no contrast.

The data in Figure 10 show the distribution of the Sound–sense variable for the three parameters: we did not introduce variables for vowels and voicing which are, however, present in the same table and allow us to evaluate the correlation between ATF and sound, which as can be seen below is negative for both *Judgement* and *Affect*:

1. Correlation between Vowels and *Judgement*: $-0.1254$;
2. Correlation between Voicing and *Judgement*: $-0.1468$;
3. Correlation between Vowels and *Affect*: $-0.08859$;
4. Correlation between Voicing and *Affect*: $-0.01346$;
5. Correlation between Judgement and *Affect*: $-0.1376$;
6. Correlation between *Affect* and *Appraisal*: $-0.0351$.

Correlations of sound data with Appraisal are on the contrary both positive. If we consider now the remaining 65 sonnets which have been classified by ATF with contrast,

we obtain a different picture. In this case, we have separated each class and projected them with sound data, Vowels and Voicing in the following three diagrams.

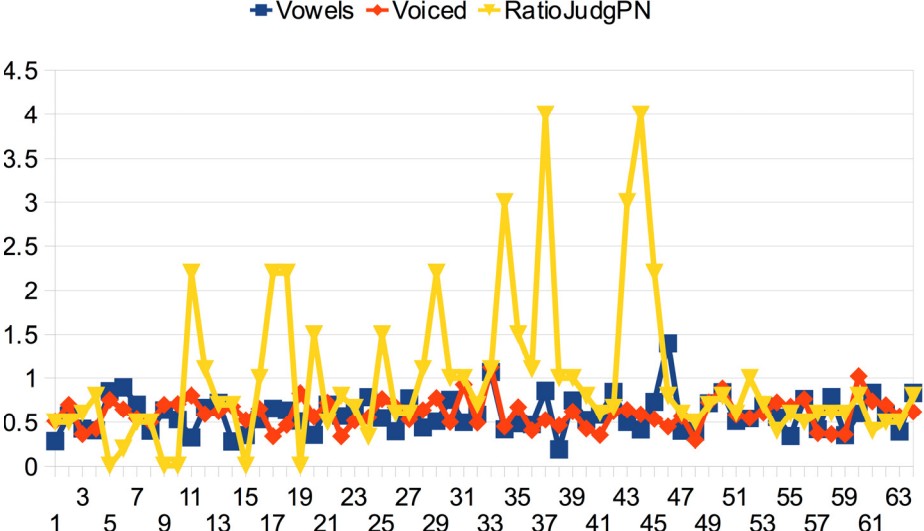

**Figure 10.** Distribution of 65 sonnets classified as Judgements with contrast and their sound data.

All correlation measures with *Judgements* are negative:

Correlation between Vowels and *Judgements*: −0.0594;
Correlation between Voicing and *Judgements*: −0.0677;
Correlation between *Judgement* and *Affect*: −0.0439;
Correlation between *Judgement* and *Appraisal*: −0.0522.

In Figure 11 below we use again sound data and the second parameter Affect:

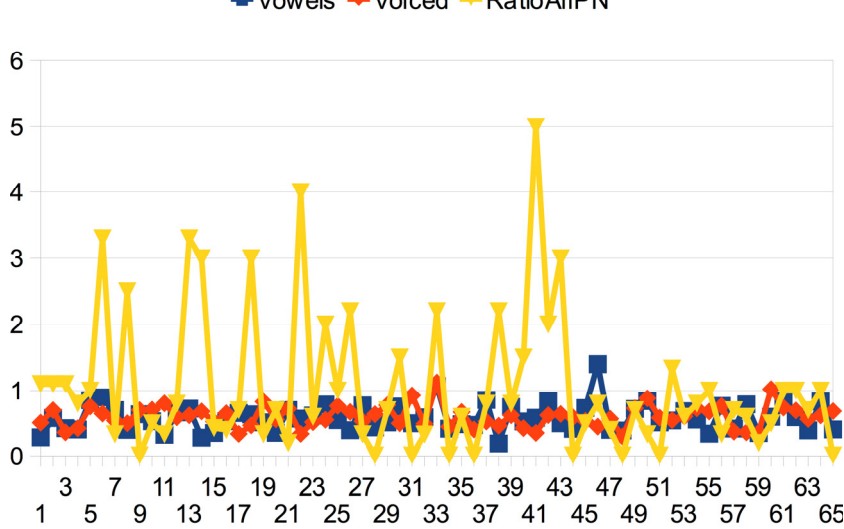

**Figure 11.** Distribution of 65 sonnets classified by ATF as Affect with contrast and their sound data.

Correlation data for *Affect* are only partly negative:
Correlation between Vowels and *Affect*: 0.09;
Correlation between Voicing and *Affect*: −0.1435;
Correlation between *Affect* and *Appraisal*: 0.2594.

Finally in Figure 12 we project sound data with Appraisal parameters:

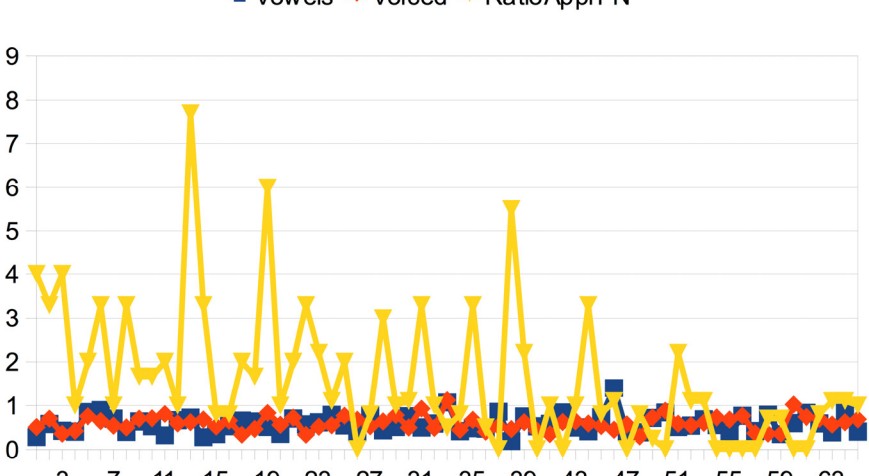

**Figure 12.** Distribution of 65 sonnets classified by ATF as Appraisal with contrast and their sound data.

Eventually, the correlations for *Appraisal* are also both negative:

Correlation between Vowels and *Appraisal*: −0.2068;
Correlation between Voicing and *Appraisal*: −0.0103.

Now the only positive correlations are the ones shown by *Affect* with Vowels and with *Appraisal*; the remaining correlations are all negative. The subdivision operated now using our manual classification with ATF seems more consistent than the one made before using the critics' evaluation. As a first comment, these data confirm our previous evaluation made on the basis of sentiment analysis, i.e., the sonnets are mainly disharmonic due to Shakespeare's intention to produce ironic effects on the audience. Here below is the list of the 89 sonnets classified by our manual ATF labeling as having no contrast:

**List of 89 sonnets manually classified by ATF as having no contrast.**

| |
|---|
| 2 6 11 13 14 16 17 20 22 23 24 26 27 30 37 38 39 40 43 44 45 46 47 51 54 56 59 62 63 64 65 66 68 71 72 74 75 76 77 78 79 81 83 84 85 87 89 90 91 96 98 99 100 101 102 103 104 105 106 107 108 109 110 111 112 113 114 115 117 118 119 122 123 124 125 126 128 130 135 136 141 145 146 147 148 149 150 |

**List of 65 sonnets manually classified by ATF as having contrast.**

| |
|---|
| 1 3 4 5 7 8 9 10 12 15 18 19 21 25 28 29 31 32 33 34 35 36 41 42 48 49 50 53 57 58 60 61 67 69 70 73 80 82 86 88 92 93 94 95 97 116 120 121 127 129 131 132 133 134 137 138 139 140 142 143 144 151 152 153 154 |

Comparing the "contrast" criterion with the sentiment-based classification is not possible; however, the "contrast" group of sonnets is included in majority by the "negatively" marked sonnets, with the exception of 16 sonnets which are the following ones:

| |
|---|
| 30 55 60 62 63 66 71 103 107 112 124 126 127 146 148 149 |

What these sonnets have in common is an identical number of *Appraisal Positive/Negative* feature (28), a high number of *Affect Negative* feature (38) vs. *Positive* ones (11), and the relatively lowest number of *Judgement Negative* features (10) vs. *Positive* ones (18). In other words, by decomposing *Negative Polarity* items into three classes, we managed to show the weakness of sentiment analysis, where Negativity is a cover-all class. Overall, the

sonnets contain a majority of positively or neutrally evaluated sonnets in both sentiment and appraisal analysis, and a minority of negatively evaluated sonnets: the SSH is, however, mostly a disharmony.

*3.3. Sound and Harmony in the Poetry of Francis Webb*

In this section, I will presents the results obtained from the analysis of the poetry by Francis Webb, who is regarded by many critics among the best English poets of the last century—and differently from Shakespeare, he never uses ironic attitudes. All the poems I will be using are taken from the Collected Poems edited by Toby Davidson [39].

I will introduce a type of graphical maps highlighting differences using colours associated with sound and sense (see [11]). The representation of the proposed harmony between sense and sound will be cast on the graphical space as follows:

- Class A:

Negatively harmonic poems, mainly negatively marked poems on the left. Either the sounds or the sentiment are in majority negative, or both the sounds and the sentiment are negative.

- Class C:

Positively harmonic poems, mainly positively marked poems on the right. Either the sounds or the sentiment are in majority positive, or both the sounds and the sentiment are positive.

- Class B:

Disharmonic ones in the middle. The sounds and the sentiment have opposite values and either one or the other have values below a given threshold.

In addition to the evaluation of positive/negative values, we consider the two parameters we already computed related to Metrical Length and Rhyming Scheme that we add together and use for its 10% added value to compensate for poetic relevant features. On the basis of poetic devices analyzed by *SPARSAR*, a list of 14 poems is considered as deviant, and they are the following: *A Sunrise, The Gunner, The Explorer's Wife, For My Grandfather, Idyll, Middle Harbour, Politician, To a Poet, The Captain of the Oberon, Palace of Dreams, The Room, Vancouver by Rail, Henry Lawson,* and *Achilles and the Woman*. In Figure 13 we show the first map of sense–sound evaluation where the split of the "deviants" poems appears clearly:

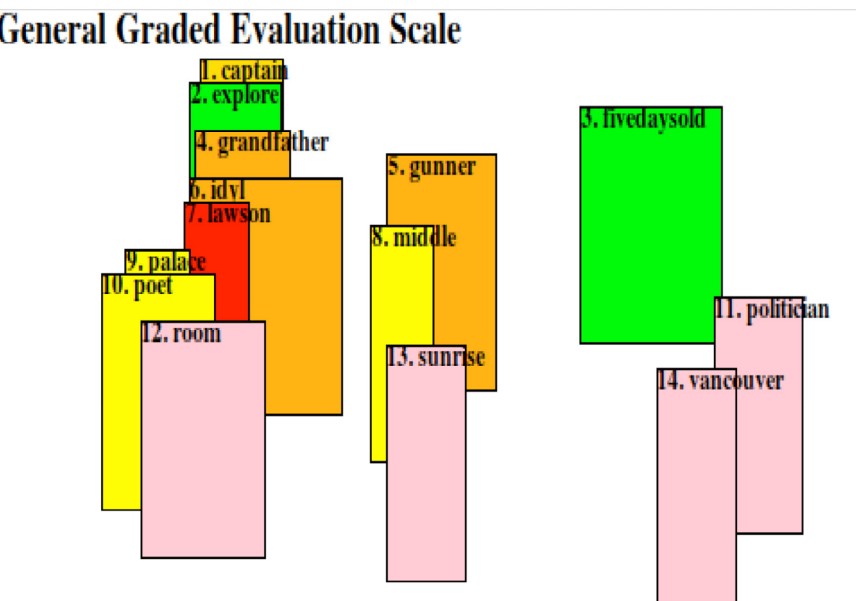

**Figure 13.** Poems considered as deviants evaluated for their degree of sense/sound harmony.

The poem that best represents balanced positive values is *Five Days Old* and this may be deduced by the presence of the largest box positioned on the right hand side. Overall, the figure shows which poems achieved harmonic values and positions positives on the right and negative on the left sides, and then in the middle disharmonic ones. As clearly appears, "Five Days Old", "Politician" and "Vancouver by Rail" are the three poems computed as endowed with positive harmony, while the remaining poems are either characterized as strongly negative—"Poet", "Palace of Dreams" and "The Room"—or just negative, "The Captain of the Oberon", The Explorer's Wife", "For My Grandfather", "Idyl", and "Henry Lawson". Finally, the last three poems positioned in the centre left are disharmonic, "The Gunner", "Middle Harbour", and "A Sunrise", where disharmonic means that the parameters of sounds are in opposition to those of sense. Slight variations in the position are determined by the contribution of parameters computed from poetic devices as said above. Disharmony as will be discussed further on might be regarded as a choice by the poet with the intended aim to reconcile the opposites in the poem.

The choice of these 14 poems includes poetry written at the beginning of the career, i.e., included in the *Early Poems*—A Sunrise, Palace of Dreams, To a Poet, Idyll, Middle Harbour, and Vancouver by Rail—two poems from *A Drum for Ben Boyd*; Politician, The Captain of the Oberon—five poems from *Leichhardt in Theatre*—The Room, The Explorer's Wife, For My Grandfather, The Gunner, Henry Lawson—and finally, one poem from *Birthday*, Achilles and the Woman, and one poem from *Socrates*, Five Days Old. In what follows, at first, I will show small groups of poems taken from different periods in Webb's poetic production and discuss them separately, rather than conflating them all in a single image. In fact, at the end of this section, I will show a bigger picture where I analysed 87 poems together, resulting in two big figures. Now, I will back to the second experiment where I collected and analyzed the following poems,

*Early Poems*—Idyll, The Mountains, Vancouver by Rail, A Tip for Saturday, This Runner
*Leichhardt in Theatre*—Melville at Woods Hole, For Ethel, On First Hearing a Cuckoo
*Poems 1950–52*—The Runner, Nuriootpa
*Birthday*—Ball's Head Again, The Song of a New Australian
*Socrates*—The Yellowhammer
*The Ghost of the Cock*—Ward Two and the Kookaburra
Unfinished Works—Episode, Untitled
In Figure 14 I show their distribution in the three separate rows:

**Figure 14.** Sixteen poems from different periods of Webb's poetic production computed for their Sense/Sound Harmony.

Here again, it is important to notice the majority of the poems positioned on the left hand side are thus analyzed as possessing negative harmony and only three poems on the right hand side, one of which is the unfinished "Untitled". And then, in the middle, there is a small number of disharmonic poems, or we could call them poems in which there were conflicting forces contributing to the overall meaning intended by the poem. Also take into account the dimension of the box which signals the major or minor contribution of the overall parameters computed as discussed in previous section, of all the linguistic and poetic features contained in the poem, but measured on the basis of their minor or major dispersion using standard deviation. In the following group, I added more poems from later work, which were computer mainly as positive:

*Birthday*—Hopkins and Foster's Dam
*Socrates*—A Death at Winson Green, Eyre All Alone, Bells of St Peter Mancroft
*The Ghost of the Cock*—Around Costessey, Nessun Dorma
*Late Poems 1969–73*—Lament for St Maria Goretti, St Therese and the Child

As showns before, also in Figure 15 the poems are positioned in three separate rows according to their overall sentiment:

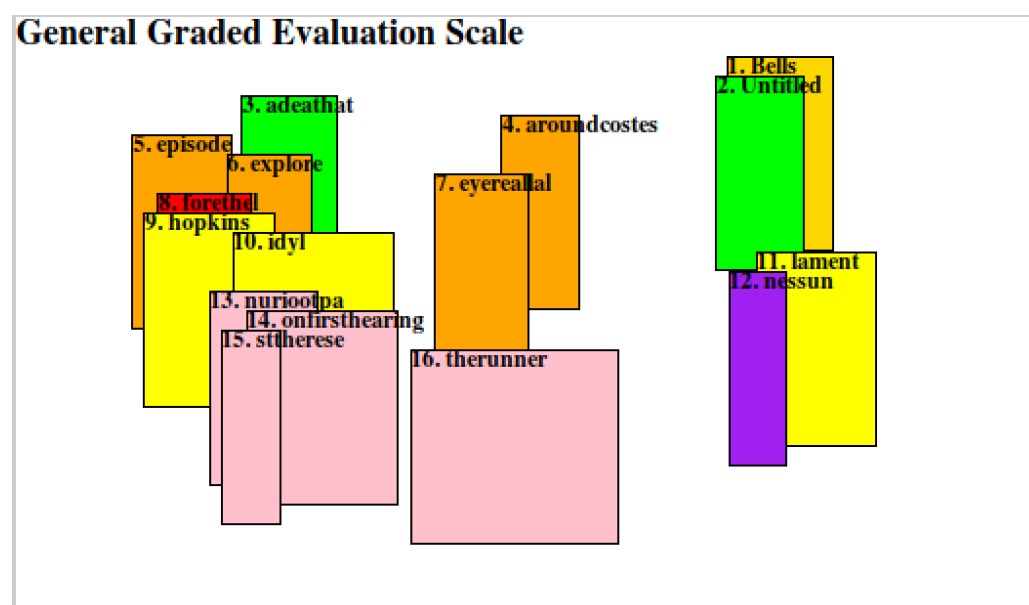

**Figure 15.** Sixteen poems taken mainly from late poetic production computed for their sense/sound harmony.

In Figure 16, I will now show a bigger picture containing 50 poems, where we can see again the great majority of them being positioned on the left hand side. The positive side is enriched by "Moonlight" from *Early Poems*, and "Song of the Brain" from *Socrates*, and the middle disharmonic list now counts 16 poems.

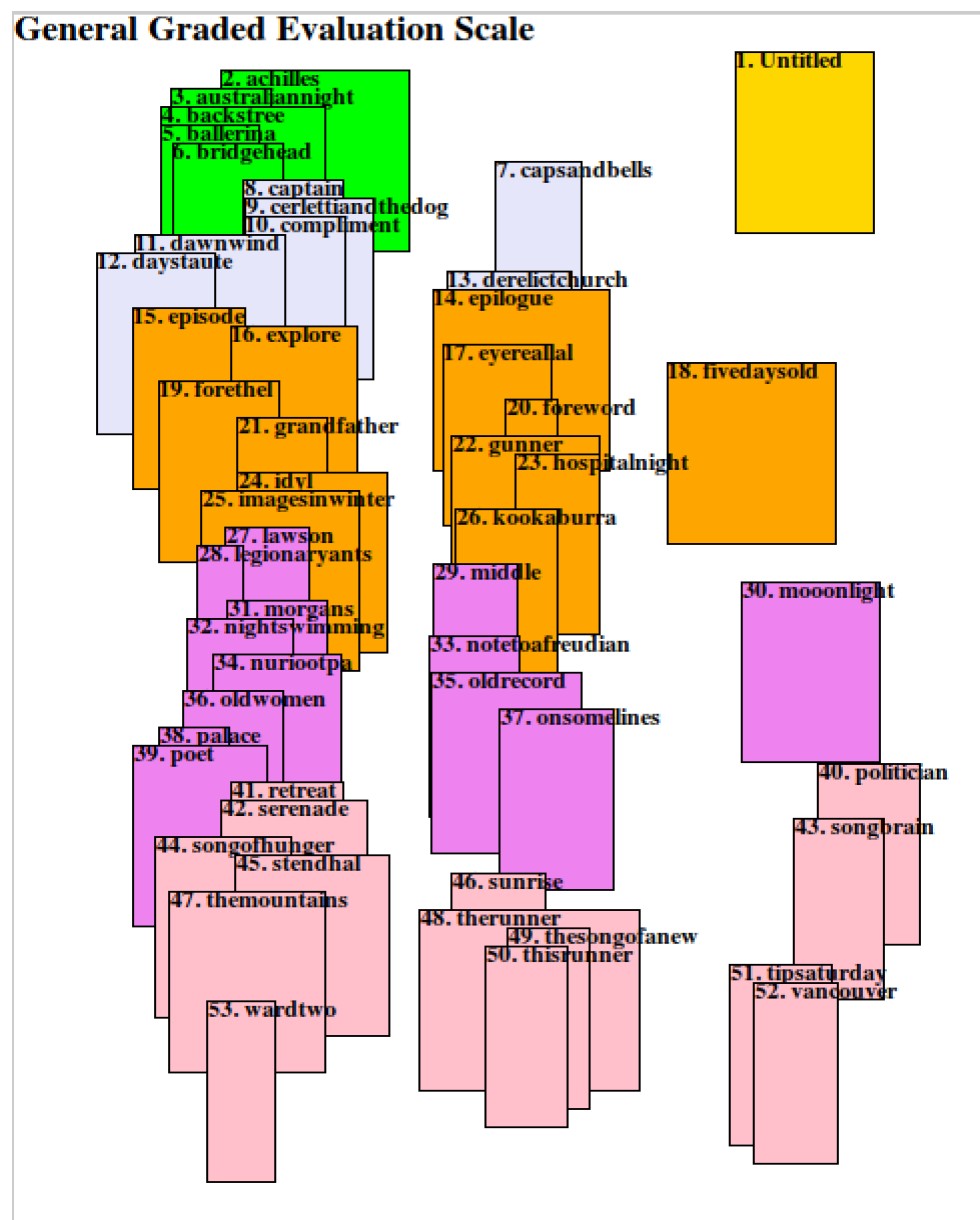

**Figure 16.** Fifty poems computed by sense/sound harmony.

So, we can safely say that the great majority of Webb's poems contain a negative harmony. This is further confirmed by the following Figure 17, which represents the analysis of 87 poems. I decided not to increase the number of poems up to 130 as was the case with the APSA system simply because otherwise the image becomes too difficult to read and poems' labels will be too cluttered together.

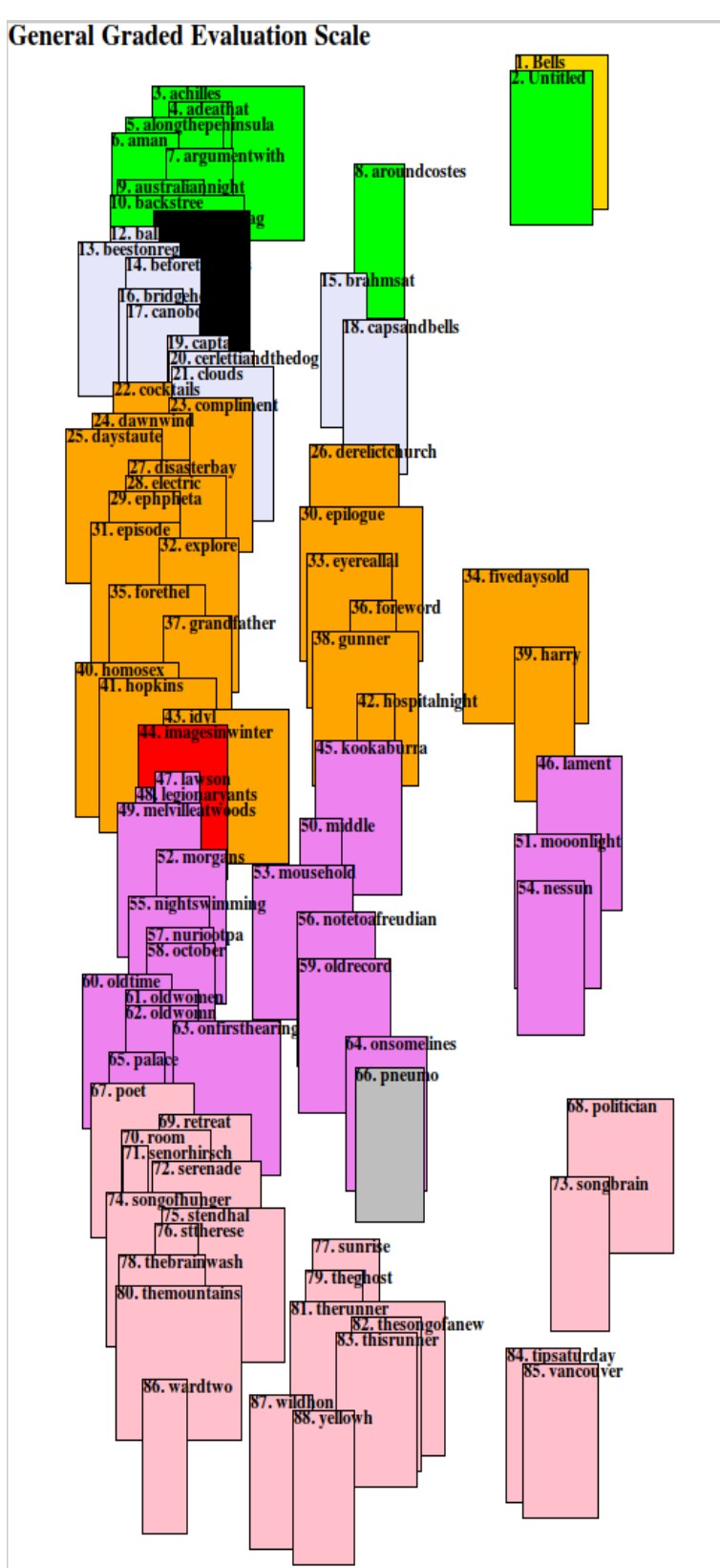

**Figure 17.** Sound–sense harmony in Webb's 87 poems.

## 4. Discussion

As now appears more clearly, the sound–sense harmony poses strict requirements on the execution of the overall experiment, which is composed of a first part dedicated

to sound harmony, thus deriving the poet's major or minor intention to fill completely the harmonic scheme with the four classes of sounds available. For this first part of the experiment, the paper has been mainly concentrated on Shakespeare's sonnets, which require a much harder level of elaboration in order to complete the sound–sense harmony experiment due to the presence of rhyme violations. As the data presented have extensively shown, sounds in Shakespeare's sonnets are mainly distributed in the four classes and the three main classes; only a few sonnets have two classes and only one sonnet has one single class. The distribution is not casual as discussed above and responds to requirements imposed by the contents. In order to obtain such an important but preliminary result, all rhyming pairs had to undergo a filtering check to evaluate their role in the overall rhyming scheme of the sonnet. In case of rhyme violation, the lexicon would have to be checked and the appropriate phonetic variation inserted.

We have then shown that the sound–sense relation may represent similar but distinct situations: in case of disharmony, we may be in presence of ironic/sarcastic expressions, as happens in Shakespeare's sonnets. This is derived from the data: as shown above, the correlation has a negative trend, meaning that the two main variables—the ones defining the behaviour of the sound patterns, and the other the behaviour of the sense, in this case the sentiment pattern—diverge and move in opposite directions. On the contrary, in the case of Webb's poetry, the contrast—when present—represents his need to encompass the opposites in life and this is testified by the frequent use of oxymora and by his condition of outcast rejected by society. Data for Webb show a great agreement in negatively marked sound–sense harmony and a much reduced agreement for positively marked data. Webb has lived half of his life in psychiatric hospitals rejected by the people who knew him, and was only accepted as a poet.

The use of two sense-related approaches has allowed us to differentiate what sentiment analysis reduced to two parameters. With the Appraisal Theory Framework, we thus managed to better specify the nature of negative sentiment using more fine-grained distinctions derived from the tri-partite subdivision of Attitude into Judgement, Appraisal and Affect. The data confirmed the previous analysis but allowed a further distinction of negatively marked sonnets into sarcastic vs. ironic.

The approach has been proven general enough to encompass poets embodying the widest possible gap from the cultural, linguistic and poetic point of view. Current DNNs are unable to cope with this task which is highly complex. It requires a sequence of carefully wrought processes in order to produce a final evaluation: in particular, the first task that is problematic for AI systems like ChatGPT is an as faithful as possible phonetic transcription of each poem. When asked to produce one such transcription, ChatGPT carried it out using IPA symbols, but as for the ARPAbet version, the result was a disaster. Word stress was assigned correctly only for a 75% of the words. The reason for this situation is very simple: dictionaries for DNN models number over one million distinct word forms and there is no resource available which counts more than 200,000 fully transcribed entries. The solution is to provide rule-based algorithms but we know that DNNs are just the opposite. They are unable to generalize what they might have learnt from a dictionary to new unseen word forms [40]. In addition, transcribing in another language—like Italian—has resulted in a complete failure. And phonetic transcription is just the first step in the pipeline of modules which are responsible for the final evaluation, as the previous section has clarified.

## 5. Conclusions

In this article, we have proposed a totally new technique to assess and appreciate poetry, the *algorithm for Sound–Sense harmony (ASSH)*. In order to evaluate poetry, we associated the phonetic image of a poem as derived from stressed syllables of rhyming words with the computed semantic and pragmatic meaning of the clauses contained in the poem. Meaning is represented by so-called "sentiment analysis" in a first approach and then by the "appraisal theory framework" in a second approach, which has offered a more fine-grained picture of the contents of each poem. We tested the technique with

the work of two famous poets, Shakespeare—an Elizabethan poet—and Francis Webb, a contemporary poet. The results obtained show the possibility to reclassify ASSH into two subcategories: **disharmony** and **harmony**, where the majority of Shakespeare's sonnets belong to the first and Webb's poetry—and as I assume the majority of current poetry—to the second. Disharmony is characterized by the presence of a marked opposition between classes—both phonetically and semantically; on the contrary, harmony is characterized by a convergence of sound and sense in the two possible nuances, negative and positive.

The data from Shakespeare's sonnets have been analyzed by usual methods with graphic charts; in the case of Webb, a new methodology has been proposed, by projecting on a graphic space the image of a poem based on its parameters, in a three dimensional manner. This is performed by drawing a coloured box representing each poem which can vary its shape according to its relevance, while its position varies according to the overall semantic parameters computed. The position of the box is assigned on one of the three sides into which the graphic space is organized: left for negatively marked harmonic poems, center for disharmonic ones, and right for positively marked harmonic poems. Boxes may vary slightly their position in one of the sides assigned according to their parameters. Differently from the results obtained for Shakespeare's sonnets, Webb's poetry—we tested the system with 100 of the most important poems—is thus characterized by a majority of poems positioned on the left, i.e., possessing negatively marked parameters for SSH. Finally, disharmony has at least two possible interpretations: in the case of Shakespeare, it represents an ironic/sarcastic mood, while in Webb's poetry, it is the result of the internal struggle for psychic survival. The method has thus been shown to be most general and applicable to any type of poetry characterizing the poet's personality by ASSH's deep analysis of the explicit and implicit contents of her/his poetic work.

**Supplementary Materials:** The following supporting information can be downloaded at: https://www.mdpi.com/article/10.3390/info14100576/s1.

**Funding:** This research received no external funding.

**Data Availability Statement:** We make available data of the complete analysis of the 154 sonnets and of 100 Webb's poems used in his section. Data is contained within supplementary material.

**Acknowledgments:** The ATF classification task has been carried out partly by Nicolò Busetto, co-author of a number of papers describing the work done. Thanks to two anonymous reviewers for the stimulating and inspiring comments that allowed me to improve the paper.

**Conflicts of Interest:** The author declares no conflict of interest.

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
