# Peer review of "Computing the Sound–Sense Harmony: A Case Study of William Shakespeare’s Sonnets and Francis Webb’s Most Popular Poems"

_information, doi:10.3390/info14100576_

Round 1

Reviewer 1 Report

The topic presented in the article is interesting, but the presentation of the content is somewhat chaotic. There is a lot of information, but presented in a way that is unclear to the reader.

It tries to cover too much: detection of irony, sentiment analysis, appraisal theory, analysis of poetry (sound-meaning relationship). Too many aspects are mixed in the article without an order that allows the reader to appreciate the author's contribution.

The title does not reflect the content of the article.

The abstract is unclear. There are too many keywords and they are too long.

The article does not present a state of the art that would allow a comparison of what is proposed with other works in the same field of study. Reference is made to various theories and fields, such as sentiment analysis, appraisal theory, irony analysis, etc., but no clear theoretical framework is provided to understand the contributions made.

The introduction does not use any bibliography to contextualise what is claimed. For example, it makes reference to "traditional judgements", but no reference is made to the literature in which these traditional judgements are presented. Current DNNs are mentioned, but there is no reference to them. Sounds are related to feelings, but no explanation is given as to why these correlations are made.

In the presentation of the method, SPARSAR is presented in an unclear way. It refers to the use of the word "expressivity" referring to many levels of syntactic-semantic intervention, but without explaining the reasons for this choice. It talks about the use of WordNet and "other similar repositories" without specifying which ones. In the presentation of the phonetic module, it is not clear why it does not make use of phonetic transcription tools that could solve some of the problems it refers to.

The results indicate that it assumes a relationship between sound and meaning that allows each class to be related to different emotions and sentiments, but it does not explain on what basis this assumption is made.

I do not see the point of the section commenting on Crystal's point of view.

The discussion section is excessively brief and does not relate what is presented in this article to other work along the same lines.

There is no section on conclusions.

Acronyms are used throughout the paper without specifying their meaning. There are writing errors. The bibliography is not cited uniformly.

There are writing errors (incomplete sentences, errors with verbs...)

Author Response

Thanks to the anonymous reviewer for the stimulating and inspiring comments: they have been very helpful and I cited it in the acknowledgements. The impression one gets from the comments is that they come from a person who is not fully acquainted with the contents of the scientific domain of the paper. There are ten questions posed and I will answer to each one here below:
- first comment regarding the reference to "traditional judgements" which according to the reviewer are missing, but in fact they are at pag. 35, beginning of the section dedicated to the second poet, Francis Webb. Maybe this was too far ahead to be read. But thanks anyway, because now that page has been moved to the front of the article, in the Introduction.

- second comment or better series of comments regarding the presence of too many aspects "without an order", and "no clear theoretical framework is provided" to understand. But the question is simply this: there is no state of art regarding the topic of the article, i.e. "the sound-sense harmony", because this is the first time that this topic is being scientifically studied. And the reason is again very simple: the topic is TOO COMPLEX for any attempt to study it scientifically. As you might have noted, it requires a phonologically motivated classification of stressed rhyming syllables which is then interpreted according to "traditional judgements"; but then it requires many modules to activate deep linguistic analysis at different levels of semantic and pragmatic imports. Sentiment, irony, appraisal, are all semantically based but they require non literal interpretation. And again thanks to the reviewer, I now have a better grasp of what is more relevant for the Discussion which has now been improved.

- third, the comment of "current DNNs" mentioned in the Introduction has a precise referent, ChatGPT.

- fourth, "sounds are related to feelings" and the reviewer wants to know "why these correlations are made", well again I refer to traditional judgements. But clearly if a person does not understand the relation intervening between SOUNDS (in the line of a poem) and FEELINGS then it could be very hard to explain - something has been tried by Tsur, see the bibliography. The same point is raised below questioning the relationship between SOUND and MEANING: in this case the assumption is not made a priori but it is derived heuristically and empirically from the data coming from the experiments. This is the content of the article and this is why the paper is important in its uniqueness.

- fifth, improving abstract and keywords, I have done it, but the title is fine when the content is understood.

- sixth, "I do not see the point of the section commenting on Crystal's", I take this comment as a consequence of the lack of understanding of the content of the paper. The importance of appropriate rhyming and rhythm is indisputably at the basis of the whole experiment, and Crystal's publications and video mess up all the phonetic representation. Since Crystal is regarded a popular - even if not properly scientifically speaking - reference for the history of the English language, I included that section. I added a short sentence to motivate the presence of the section.

- seventh, presenting SPARSAR may require more than 40 additional pages, I just mentioned the main features. References are given to deepen the presentation of the system.

- eighth, the same applies to explaining the use of the term "expressivity" which has been thoroughly presented in previous papers with all needed details, but again it will require another additional 20 pages to deepen the presentation of the system.

- ninth, the interesting note on the "phonetic module", and the comment "it is not clear why it does not make use of phonetic transcription tools that could solve some of the problems it refers to". Here the reference to phonetic transcription tools that could solve some of the problems has been done without adding the actual reference the reviewer had in mind. As far as I know, there is no phonetic transcription tool able to modify the current phonetic transcription in the one needed by the sonnets and used by Shakespeare in the Elizabethan times. These analyses have already been presented in international conferences and no previous phonetic tool has been ever mentioned. But of course, if there is one such tool, I'd love to use it instead of my complex algorithm.

- tenth, yes pity that the reviewer mentions the possibility of improvements without indicating where and how: "there are writing errors (incomplete sentences, errors with verbs...)". I have been referee and reviewer in the last 30 years for international conferences, journals, international research institutions and one of the thing that I've always done is including a part of the report where typos and errors are clearly individuated in the text. This allows the paper to be improved. So what I usually tell my collaborators when they are given the same task, is to carefully read the paper and note separately every linguistic and non-linguistic element that needs mending, and to do that while reading it the first time.
Finally I added a conclusion.

Reviewer 2 Report

Is this paper, the author aims to compute the sound-sense harmony in poetic works of famous poets such as Shakespeare and Webb.

Given the chosen datasets, the results look good enough to verify the author's hypothesis, but main concern is the value of this work and how it could be useful to others. Some significant questions that one could also ask are:

- Why choosing Shakespeare and Webb specifically?

- How can those insights generalize to other poetry?

- Can the author put the works of Shakespeare and Webb in the same bin or at the same level the way it was done in this paper? The author seems to treat them as one type of poetry and having the similar importance.

Concerning the writing, the paper is well-written, but includes many long written explanations that could be better conveyed to the reader in terms of graphics, especially in Section 2.

Concerning Figures, please include PDF exports of the graphics so that they don't get pixelated when zooming. Some graphics are not clear even without Zooming.

The English writing is fine.

Author Response

Thanks for the review. Here below the answers to the questions posed.
- first, "why choosing Shakespeare and Webb", the answer is dual: first of all, because the Sonnets constitute a statistically significant dataset, and the same applies to Webb's poetry. The second and more obvious answer is that I had already been working on both poets in the past and so I had all the data needed which now allow me to put forward the new theory.

- second, "How can these insights generalise to other poetry", well this is harder to know. In fact, the experiment covers two possible cases, Shakespeare mainly using sound-sense harmony to enhance ironic effects, while Webb uses the same combination to make patent his internal psychological drama, by distributing harmony in three subcategories, negative, positive and balanced or neutral. The semantic and pragmatic realm of non-literal language is vast and requires subtle intuitions which may be detected in the poetry alone. 

- in the submission, I have included all figures separately in a zip file all converted into PDF format. I improved on one of the figures that didn't look clear enough.

Reviewer 3 Report

The abbreviation SSH is unfortunate because it is widely used for Secure Shell, a network communications protocol that allows encrypted communication between two computers.

The article discusses the IT-interpretable description of the analysis of the poems at great length. This stage is actually a kind of feature engineering, which is fundamental in machine learning. The discussed topic is very special, at first it seems distant compared to the usual neural network applications.

The procedure described in the article associates the phonetic image of the poem from the stressed syllables of the rhyming words with the semantic and pragmatic meaning of the clauses in the poem for sound-sensual harmony. The meaning is partly represented by sentiment analysis and partly by the theoretical framework of evaluation, the latter offering a finer picture of the content of each poem.

In addition to the literary application, the method described in the article can also be used with appropriate changes to analyze everyday speech and to better understand what each speaker has to say.

The shortcoming of the article is that it does not pay enough attention to the applicability of the method to other poems. A short (one or two paragraphs) description would be appropriate and interesting for the reader.   The article is a well-structured work that describes the topic in detail, but not at length.

Author Response

Thanks for the comments. I did the following changes:
- modified SSH into ASSH (Algorithm for Sound Sense Harmony)
- introduced short sentences in the abstract, in the discussion and in the conclusion highlighting the general relevance of the approach 

Round 2

Reviewer 1 Report

Although the abstract has been slightly improved, there is still a problem with it: a "second poet" is referred to but not named. This makes that part of the abstract uninterpretable if the title is not available. Add the name of the poet.

Keywords are still not "words", they are syntagms/phrases. Use keywords.

The introduction has improved with the inclusion of some reference.

The discussion has also improved and conclusions have been included.

The references needs to be revised to make them uniform. Sometimes the author gives the author's full surname and first name (Crystal David); sometimes full surname and abbreviated first name (e.g. Ingham, R., & Ingham, M.), sometimes abbreviated first name and then surname (e.g. J. Martin and P.R. White); sometimes surname and then abbreviated first name (e.g. McGuire, P. C.); sometimes full first name and then full surname (Michele Stingo and Rodolfo Delmonte). In short, there is no uniformity in the citation.

I suggest an orthotypographical revision. I am not going to list the typos here, but as an example, see the acknowledgements: "anonymous reviwer".

The quality and size of the figures should also be reviewed.

I suggest an orthotypographical revision. 

Author Response

Again thanks for the useful comments. I did the following:
- changed "second poet" with Francis Webb
- improved Keywords by shortening them

Reviewer 2 Report

The author still hasn't convinced me about the relevance of this work and its value to others.

Author Response

Sorry, maybe the latest revision will do the miracle!!